# ADAPTIVE FIDELITY-DRIVEN RECONSTRUCTION (AFR): A REALISTIC THREAT MODEL FOR SPECTRAL EMBEDDING LEAKAGE

## ABSTRACT

The exchange of structural representations in Federated Graph Learning (FGL) creates a potent channel for privacy leakage. While theoretical graph reconstruction is possible, existing attack models are brittle as they hinge on an unrealistic assumption: perfect, noise-free local data. This paper elevates that theoretical threat to a practical reality. We introduce AFR (Adaptive Fidelity-driven Reconstruction), a robust new attack model that abandons idealized assumptions. Instead of assuming data quality, AFR actively measures and exploits it. The algorithm first quantifies the reliability of each local patch via a novel fidelity score, combining a spectral signal-to-noise ratio with structural entropy. This score then guides a robust assembly process that uses RANSAC-Procrustes to tolerate outliers and adaptive stitching criteria to manage uncertainty. Instead of a single, perfect graph, AFR recovers large, high-fidelity, and internally consistent islands from the most trustworthy data. Experiments on the LoGraB benchmark show that AFR successfully reconstructs significant topology in challenging, noisy regimes where idealized models fail completely. Our work thus promotes spectral leakage from a theoretical possibility to a practical and potent threat. Our source code is anonymously available at: https://anonymous.4open.science/r/AFR-ICLR-submission.

## 1 INTRODUCTION

Federated Learning (FL) promises a future of privacy-preserving machine learning, where models are trained collaboratively without sharing raw data. This promise rests on a core premise: that exchanging only intermediate representations, like gradients or embeddings, is safe. But this premise is fragile. The intermediate signals themselves can inadvertently leak sensitive information. In Federated Graph Learning (FGL), this challenge is amplified. The primary asset to protect is not just the node features, but the local graph topology, which can encode sensitive relationships. It is important to distinguish our threat model from model-level attacks (gradient inversion). AFR targets a data-level vulnerability: the passive leakage of local spectral embeddings. Unlike gradients which are transient, these embeddings are often shared as static artifacts for alignment or clustering, making them a persistent privacy risk independent of the training process. Recent comprehensive evaluations have confirmed the severity of this threat through various reconstruction and inference attacks (Chen et al., 2024). This vulnerability extends beyond topology, with emerging work showing leakage of sensitive node properties (Liu et al., 2025) and even local label distributions from shared representations. Spectral embeddings, a common currency for exchanging this structural information, create the most potent passive leakage channel. Unlike active attacks that risk detection, a passive attack requires only observing a single, honestly-shared artifact. The vulnerability, therefore, lies not in the federated protocol, but in the intrinsic properties of the representation itself. This threat is more fundamental, challenging core assumptions about the safety of distributed graph learning.

To understand the practical threat of spectral leakage, we first consider a best-case scenario. We construct an idealized stitching model, a baseline that proves local spectral embeddings can, in principle, be stitched together for exact global reconstruction. This confirms that simple data partitioning is not an effective defense. However, the power of this model depends on a set of unrealistic conditions. It demands a near-unattainable level of data perfection, requiring both: (1) that every local patch exhibits a significant spectral gap for noiseless inversion and (2) that

every adjacent patch overlap is large and well-connected for perfect alignment. This is the "curse of perfection". The reconstruction is a deterministic cascade; a single patch that violates these conditions introduces an initial error that propagates and catastrophically corrupts the entire global assembly. The approach is thus fundamentally brittle. This brittleness reveals a critical gap between theoretical possibility and practical threat, motivating the central question of our work: Can an adversary reconstruct a meaningful topology when the leaked information is inevitably noisy, incomplete, and of heterogeneous quality?

Our answer is **AFR** (Adaptive Fidelity-driven Reconstruction), an algorithm built on a new philosophy: Instead of assuming that the data are perfect, it embraces its imperfection. AFR actively measures, quantifies, and exploits the varying quality of leaked information. First, it assigns each patch a novel fidelity score ($s_v$), combining spectral stability (SNR) and topological complexity (Structural entropy) to gauge its reliability (shown in Section 3.2). Only the most trustworthy core patches are even considered for assembly. The assembly process is then designed for failure. A robust RANSAC-Procrustes alignment handles the inevitable outliers from noisy reconstructions, and an adaptive stitching threshold demands more evidence (a larger overlap) to connect less reliable patches. This fidelity-driven approach redefines success. AFR abandons the fragile pursuit of a single global graph. Instead, its goal is to recover a group of large, internally-consistent, high-fidelity islands from the most reliable data, while intelligently isolating what cannot be recovered.

Our work makes the following contributions:

- **A practical threat model for spectral leakage**. We are the first to formalize and address spectral leakage under realistic and imperfect conditions. Our approach accounts for noise, reconstruction errors, and heterogeneity, elevating the threat from a theoretical possibility to a practical concern.
- **A computable fidelity score**. We introduce a novel fidelity score that quantifies the reliability of leaked graph patches. By combining a spectral signal-to-noise ratio (SNR) with structural entropy, this score provides a quantitative foundation for robust decision-making.
- **Adaptive Fidelity-driven Reconstruction (AFR) algorithm**. We propose AFR, an adaptive, multi-stage algorithm that uses our fidelity scores to guide a robust assembly process. Key components include an outlier-resilient RANSAC-Procrustes alignment and adaptive stitching criteria to manage uncertainty.
- **Extensive empirical validation**. We conduct experiments across a wide range of realistic, challenging scenarios generated by our evaluation protocol. Our results demonstrate that AFR successfully reconstructs significant topology in regimes where idealized prior methods fail completely, thus confirming the viability of our new threat model.

## 2 RELATED WORK

**Privacy threats in Federated Graph Learning** The premise that sharing intermediate representations in Federated Learning (FL) is safe has been systematically challenged. In particular, Deep Leakage from Gradients (DLG) demonstrated that raw training data can be recovered from shared gradients (Zhu et al., 2019; Geiping et al., 2020). Although our work shares this inversion philosophy, we target a fundamentally different leakage channel: not dynamic gradients, but static spectral embeddings of the graph structure. This vulnerability applies to any FGL scenario in which structural representations are exchanged, even if no gradients are shared. Within the taxonomy of graph privacy attacks, AFR instantiates a passive eavesdropper threat model. As summarized in the table below, this is distinct from active query-based attacks such as link inference (He et al., 2021b) or membership inference (Zhang et al., 2021), which risk detection. It also differs from malicious-client attacks. The passive model represents a more fundamental threat, as the vulnerability is an intrinsic property of the honestly-shared data artifact itself, making it virtually undetectable at the protocol level. Comprehensive empirical studies have formalized Graph Reconstruction Attacks (GRA) as a major vulnerability (Chen et al., 2024). Beyond structural recovery, sophisticated influence-based attacks have been developed to infer links with high accuracy (Wu et al., 2022; Meng et al., 2023). The severity of membership inference, which AFR's threat model is related to, has prompted the recent development of dedicated defenses, highlighting the community's focus on this problem (Zhong et al., 2025).

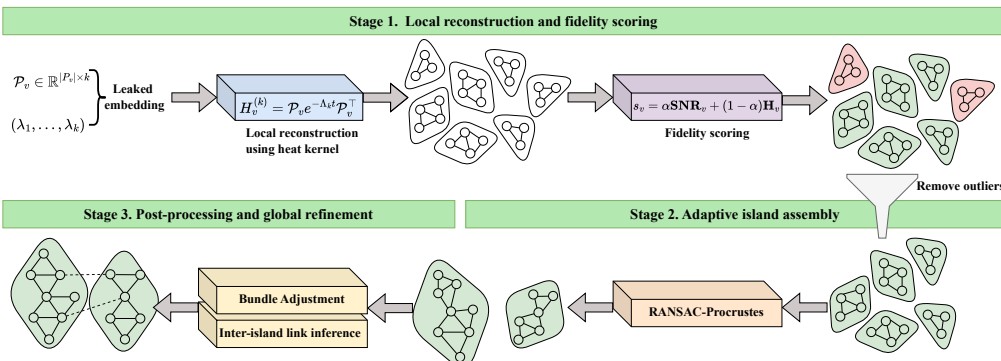

Figure 1: **Adaptive Fidelity-driven Reconstruction (AFR)**. The pipeline reconstructs graphs in three stages: local reconstruction with fidelity scoring, adaptive island assembly using RANSAC-Procrustes, and global refinement via Bundle Adjustment and inter-island link inference.

**Graph reconstruction from Local Embeddings** Our work builds upon the theoretical principle that a global graph can, under idealized conditions, be perfectly reconstructed by stitching together local spectral embeddings. The fundamental feasibility of inverting spectral representations is supported by theoretical findings demonstrating that, under certain identifiability conditions, graphs are reconstructible from their eigenspaces, even in the presence of noise (Castro et al., 2017). But this guarantee comes with a curse of perfection: it demands that every patch and every overlap be flawless. A single imperfect patch is enough to trigger a catastrophic cascade of errors, which makes the entire pipeline fundamentally brittle. AFR is designed precisely to break this curse, operating on the noisy, imperfect embeddings found in realistic settings. Conceptually, our problem can be framed as an "Inverse GAE". While a Variational Graph Auto-Encoder (Kipf & Welling, 2016) learns a mapping from a graph to an embedding, $p(\mathcal{A}|\mathcal{Z})$, our challenge is the inverse: given a collection of corrupted localized embeddings $\mathcal{P}$, we must infer a plausible graph, $p(\mathcal{A}|\mathcal{P})$.

**Robust geometric alignment** The resilience of AFR is inspired by battle-tested techniques from computer vision, where dealing with noisy and imperfect data is the norm. Standard Orthogonal Procrustes analysis is notoriously sensitive to outliers (Schönemann, 1966). To overcome this, we replace it with RANSAC-Procrustes (Fischler & Bolles, 1981), classic paradigm designed to find a reliable fit even in the presence of faulty data points. To correct for small alignment errors that accumulate into large-scale distortions, AFR employs a post-processing step analogous to Bundle Adjustment (Triggs et al., 2000), a cornerstone of structure-from-motion that jointly optimizes for global geometric consistency.

**Defenses against information leakage in FGL** The significant privacy risks highlighted by AFR and related works have spurred the development of various defense mechanisms. A prominent line of defense involves applying Differential Privacy (DP). This includes adding noise to shared parameters to provide formal privacy guarantees (Qiu et al., 2022) or perturbing the aggregation function within GNNs to protect edge-level privacy (Sajadmanesh et al., 2023). Another approach uses information-theoretic principles, such as the information bottleneck, to learn minimal, privacy-preserving subgraphs that are shared instead of the original topology (Zhang et al., 2023). Finally, hybrid methods combine techniques like Local Differential Privacy (LDP) with secure hardware such as Intel SGX to create multi-layered protection (Han et al., 2024). These defensive strategies underscore the urgent need for realistic threat models such as AFR to properly evaluate their effectiveness.

In summary, while previous work has separately explored reconstruction attacks or robust alignment, AFR is the first to synthesize these principles into a coherent framework that addresses the practical threat of leakage from imperfect, localized spectral embeddings. Our work is at the intersection of privacy in federated systems, graph reconstruction attacks, and robust geometric alignment.

## 3 ADAPTIVE FIDELITY-DRIVEN RECONSTRUCTION (AFR)

### 3.1 THEORETICAL UNDERPINNINGS

Our theoretical framework is built upon a set of formal assumptions that guide the algorithm's behavior under realistic, imperfect conditions. Instead of demanding data perfection, we assume that a subset of the local data is sufficiently reliable to initiate a robust reconstruction and that this reliability can be quantified. First, we formally define a core patch as a local subgraph that is computationally tractable, has a bounded reconstruction error from its spectral embedding, and crucially, surpasses a minimum threshold on our composite fidelity score. This score ensures that we only build upon patches that are both spectrally stable and structurally informative. Second, we establish an adaptive eligibility criterion for stitching, assuming that a pair of core patches can only be aligned if their overlap is sufficiently large and structurally sound. Critically, this required overlap size increases as the fidelity of the patches decreases, forcing the algorithm to demand stronger evidence when faced with greater uncertainty. Finally, we rely on the standard probabilistic guarantees of our alignment method, assuming that for any eligible pair, RANSAC-Procrustes can recover a near-correct relative rotation with high probability. These assumptions collectively create a principled foundation for a reconstruction algorithm that is resilient to noise and heterogeneity. The complete mathematical formulation of these assumptions is detailed in Appendix A.

### 3.2 METHODOLOGY

Let $G = (V, E)$ be an undirected graph, $|V| = n$. For every $v \in V$ we observe a local $d$-hop embedding $\mathcal{P}_v \in \mathbb{R}^{|P_v| \times k}$ of the patch $P_v \subseteq V$. Our goal is to reconstruct a set of high-fidelity islands $\{\widehat{G}_1, \ldots, \widehat{G}_r\} \subseteq G$ together with a probabilistic inter-island edge set $\widehat{E}_{\text{cross}}$.

**Stage 1. Local reconstruction and fidelity scoring**   The first stage of AFR acts as a gatekeeper. Its goal is to transform each raw, leaked spectral embedding into a structured representation and then quantify how trustworthy it is as an anchor for reconstruction.

**Local reconstruction**.   For each patch $v$, we reconstruct a local adjacency matrix $\widehat{\mathcal{A}}_v$ from its spectral embedding $\mathcal{P}_v \in \mathbb{R}^{|P_v| \times k}$ and corresponding eigenvalues $(\lambda_1, \ldots, \lambda_k)$ using the Heat kernel method. This inverts the embedding process to produce a tangible estimate of the local topology. We first form the truncated heat kernel approximation: $H_v^{(k)} = \mathcal{P}_v e^{-\Lambda_k t} \mathcal{P}_v^\top$, where $t$ is a small fixed-time parameter and $\Lambda_k = \text{diag}(\lambda_1, \ldots, \lambda_k)$. The entries of the true heat kernel exhibit a positive separation gap between the connected and disconnected nodes, allowing for accurate recovery. The local adjacency matrix $\widehat{\mathcal{A}}_v$ is then obtained by applying a threshold $\tau_v$ to the entries of $H_v^{(k)}$.

**Fidelity scoring**.   But a reconstruction is only as good as its source data. The core innovation of this stage is a multi-component fidelity score, $s_v$, designed to answer two orthogonal questions about each patch's quality:

1. **Is the signal clean?** We quantify this with the spectral **signal-to-noise ratio (SNR)**. This ratio measures the stability of the local eigenspace against truncation errors, giving us a principled measure of information quality.

$$\text{SNR}_v = \frac{\delta_v}{\delta_v + \eta_v}.$$

    where the signal $\delta_v = \lambda_{k+1}(\mathcal{L}_v) - \lambda_k(\mathcal{L}_v)$ and the noise $\eta_v = e^{-t\lambda_{k+1}(\mathcal{L}_v)}$. This formulation provides the information quality relative to the information lost by truncation.

2. **Is the signal distinctive?** We quantify this with **Structural entropy**. This metric gauges the topological complexity by using the normalized Shannon entropy ($\mathcal{E}_v$) of the patch's degree distribution.

$$\mathcal{E}_v = -\frac{1}{\log(|V_v|)} \sum_{d \in D_v} p(d) \log\big(p(d)\big),$$

    where $D_v$ is the set of unique node degrees in the patch and $p(d)$ is the empirical probability that the node has degree $d$. A patch might have a clean signal but a trivial structure (low entropy) or be complex but noisy (low SNR). A trustworthy patch needs both.

These two components are then combined into a composite fidelity score via a convex combination:

$$s_v = \alpha \mathrm{SNR}_v + (1 - \alpha)\mathcal{E}_v,$$

where $\alpha \in [0, 1]$ balances the relative importance of spectral stability and structural complexity. Finally, we filter for quality: Only patches that surpass the minimum threshold for both their fidelity score and their spectral gap are promoted to "core patches", forming the trusted input for the next stage.

**Stage 2. Adaptive island assembly**     This stage is the heart of AFR, where the trustworthy "core patches" are intelligently assembled into larger, internally consistent structural "islands".

**Data-driven prioritization**. To minimize error propagation, the assembly follows a prioritized scheme. The set of core patches $(\widehat{\mathcal{A}}_v, s_v)$ that satisfy Assumption 5 forms the input to this stage. The priority queue then determines the next best potential "stitch", based on a combination of the joint fidelity of the patch pair $f(s_v, s_w)$ and the size of their overlap $|I_{vw}|$.

**Adaptive stitching loop**. The algorithm iteratively attempts to stitch the highest-priority pair $(v, w)$ from the queue. This process involves a rigorous multi-step verification:

1. Eligibility check: First, the pair must satisfy our criterion: Their overlap must be large enough and the induced subgraph must be structurally connected. This overlap size threshold increases for lower-fidelity patches, forcing the algorithm to demand stronger evidence when faced with more uncertainty.
2. Robust alignment: If eligible, we align the pair using RANSAC-Procrustes. This choice is deliberate: It is designed to find a correct rotation even in the presence of faulty correspondences, an inevitability given reconstruction errors (see details in Algorithm 1).
3. Stitching decision: A stitch is accepted only if RANSAC finds a strong geometric consensus in a significant portion of the overlap (max_consensus_count $\geq d_{\mathrm{adaptive}}(s_v, s_w)$). This final check ensures that the stitches are based on coherent evidence, making the entire assembly process highly robust.

**Stage 3: Post-processing and global refinement**     The final stage of AFR performs two operations: refining the internal geometric consistency of each island and inferring latent connections between them.

**Intra-island refinement via Bundle Adjustment**. First, we correct for "drift". Even with robust pairwise stitching, small alignment errors can accumulate across large islands, causing subtle distortions. To fix this, we employ a global refinement step analogous to Bundle Adjustment from computer vision. We jointly optimize all pairwise rotations within an island to minimize a global geometric error, effectively "truing up" the entire structure.

$$\{\mathcal{Q}_{vw}^*\} = \mathrm{argmin}_{\{\mathcal{Q}_{vw}\}} \sum_{(v,w)\in\mathrm{stitches}(G_i)} \|\mathcal{P}_v' - \mathcal{P}_w'\mathcal{Q}_{vw}\|_F^2 ,$$

where $\mathcal{P}_v'$ and $\mathcal{P}_w'$ are the eigenvector submatrices corresponding to the overlap. This non-convex optimization problem over the manifold of orthogonal matrices can be solved using iterative methods such as Riemannian gradient descent (see details in Appendix E.5).

**Inter-island link inference via cross-voting**. Finally, we hunt for the potential links between our recovered islands. To do this, we return to the raw data and use a cross-voting mechanism. The intuition is simple: if two nodes from different islands $u$ and $w$ frequently co-occurred in the initial, low-quality patches, it constitutes strong latent evidence of a true connection. We formalize this by computing a vote count, $C(u, w)$, and if it surpasses a threshold, we infer a probabilistic edge. This allows AFR to reason about the global topology, even the parts it could not confidently reconstruct.

$$P_{inter}(u, w) = \frac{1}{1 + e^{-\beta(C(u,w)-C_0)}},$$

where $C_0$ is a vote threshold and $\beta$ controls the steepness of the probability curve.

## 3.3 Theoretical Guarantees

**Notation.** For a local patch $P_v$ with $n_v = |P_v|$ nodes, let $\mathcal{L}_v$ be its (combinatorial or normalized) Laplacian with eigenpairs $\{(\lambda_r, u_r)\}_{r \geq 1}$, ordered nondecreasingly. The truncated heat kernel is

$$H_v^{(k)}(t) = \sum_{r=1}^{k} e^{-t\lambda_r} u_r u_r^\top, \quad \text{and the residual } R_v^{(k)}(t) = H_v(t) - H_v^{(k)}(t) = \sum_{r>k} e^{-t\lambda_r} u_r u_r^\top.$$

We reconstruct $A_v$ by thresholding the entries of $H_v^{(k)}(t)$. The patch-wise spectral gap is $\delta_v = \lambda_{k+1} - \lambda_k$ and the truncation proxy is $\eta_v = \|R_v^{(k)}(t)\|_2 = e^{-t\lambda_{k+1}}$. The composite fidelity is $s_v = \alpha \cdot \mathrm{SNR}_v + (1 - \alpha) \cdot \mathcal{E}_v$ with $\mathrm{SNR}_v = \delta_v / (\delta_v + \eta_v)$ and $\mathcal{E}_v$ the structural entropy.

**Theorem 1** (Edge recovery). *Let $t > 0$ be a fixed time parameter. Suppose the true heat kernel $H_v(t)$ on $P_v$ exhibits an entry-wise separation margin between its edge and non-edge values, defined as:*

$$\gamma_t = \min_{(i,j) \in E(v)} H_v(t)_{ij} - \max_{(i,j) \notin E(v)} H_v(t)_{ij} > 0.$$

*Let the truncation error be bounded by $\eta_v = e^{-t\lambda_{k+1}}$, which corresponds to the spectral norm of the residual matrix $R_v^{(k)}(t) = H_v(t) - H_v^{(k)}(t)$. If the separation margin is greater than twice the error bound $\gamma_t > 2\eta_v$, then the edge set $E(v)$ can be recovered exactly. Specifically, there exists a non-empty interval of thresholds $\tau_v$ such that for any $\tau_v$ in this interval, setting $\widehat{E}_v = \{(i,j) | H_v^{(k)}(t)_{ij} > \tau_v, i \neq j\}$ yields $\widehat{E}_v = E_v$.*

(Proof is provided in Appendix E.1).

**Proposition 2** (Basic properties of the fidelity score). *Let the spectral signal-to-noise ratio be defined as*

$$\mathrm{SNR}_v = \frac{\delta_v}{\delta_v + \eta_v},$$

*where $\delta_v = \lambda_{k+1} - \lambda_k$ is the spectral gap and $\eta_v = e^{-t\lambda_{k+1}}$ is the truncation error bound. Let $\mathcal{E}_v$ be the normalized Shannon entropy of the patch's degree distribution, such that $\mathcal{E}_v \in [0, 1]$. The composite fidelity score is*

$$s_v = \alpha \cdot \mathrm{SNR}_v + (1 - \alpha) \cdot \mathcal{E}_v \quad \text{for a constant } \alpha \in [0, 1].$$

*The following properties hold:*

1. *$\mathrm{SNR}_v$ is bounded, i.e., $0 \leq \mathrm{SNR}_v \leq 1$. It is a strictly increasing function of the spectral gap $\delta_v$, and a strictly decreasing function of the truncation error $\eta_v$.*
2. *The composite fidelity score $s_v$ is bounded, that is, $0 \leq s_v \leq 1$.*
3. *The score $s_v$ is monotonically non-decreasing with respect to the spectral gap $\delta_v$.*

(The proof can be found in Appendix E.2).

Let $I_{vw} = P_v \cap P_w$ and $p_{vw} \in (0, 1]$ be the inlier fraction among the correspondences on $I_{vw}$. The adaptive admissibility requires $|I_{vw}| \geq d_{\mathrm{adapt}}(s_v, s_w) := k_{\mathrm{base}} + \gamma(1 - \min\{s_v, s_w\})$ and the connectivity of $G[I_{vw}]$.

**Lemma 3** (RANSAC all-inlier sample probability). *Let $I_{vw}$ be the set of correspondences identified in the overlap of two patches, of which an unknown fraction $p_{vw} \in (0, 1]$ are true inliers. Let $m$ be the minimal number of correspondences required to uniquely determine a rotation model. If the RANSAC algorithm is executed for $M$ independent iterations, where each iteration uniformly samples $m$ correspondences from $I_{vw}$, then:*

1. *The probability of drawing at least one sample consisting entirely of inliers is*

$$1 - \left(1 - p_{vw}^m\right)^M.$$

2. *Consequently, to ensure this success probability is at least $1 - \beta$ for a desired failure rate $\beta \in (0, 1)$, the number of iterations $M$ must satisfy*

$$M \geq \frac{\log(\beta)}{\log\left(1 - p_{vw}^m\right)}.$$

**Theorem 4** (Stitch soundness under adaptive evidence). *Let a pair of core patches $(v, w)$ be eligible for stitching. Assume that their overlap $I_{vw}$ contains a fraction $p_{vw} \in (0, 1]$ of true inlier correspondences, which are perturbed by zero-mean, i.i.d. sub-Gaussian noise with variance parameter $\sigma^2$. Let the RANSAC algorithm be executed for a sufficient number of iterations $M$ to ensure that an all-inlier sample is found with probability at least $1 - \beta$, as specified in Lemma 3. A stitch is accepted if and only if the size of the resulting consensus set, denoted $|C_{vw}|$, meets the adaptive threshold:*

$$|C_{vw}| \geq d_{adapt}(s_v, s_w).$$

*Under these conditions, with probability at least $1 - \beta$, the rotation $\widehat{\mathcal{Q}}_{vw}$ returned by the RANSAC–Procrustes procedure is close to the ground-truth rotation $\mathcal{Q}^*_{vw}$. Specifically, the error is bounded by*

$$\left\| \sin \Theta \big( \widehat{\mathcal{Q}}_{vw}, \mathcal{Q}^*_{vw} \big) \right\|_F \leq C \cdot \frac{\sigma}{\sqrt{d_{adapt}(s_v, s_w)}},$$

*where $C$ is a constant depending on the geometric properties of the true point configuration, and $\| \sin \Theta(A, B) \|_F$ is the canonical distance metric on the special orthogonal group. This result shows that a lower fidelity score, which increases $d_{adapt}$, enforces a more stringent error bound on the accepted alignment.*

## 4 EXPERIMENTS

To rigorously evaluate AFR's performance, we first establish a comprehensive evaluation protocol, as no standard benchmark exists for this specific task. Although benchmarks such as FedGraphNN (He et al., 2021a) exist for FGL algorithms, they are not designed to systematically test graph reconstruction against fragmented, noisy, and localized spectral leakage. To fill this gap, our protocol defines a procedure for generating a suite of challenging benchmark instances. We refer to this framework as LoGraB (Local Graph Benchmark) and will release its generation code and datasets upon acceptance to facilitate future research.

### 4.1 CORE BENCHMARK EVALUATION

Our initial evaluation rigorously adheres to the abovementioned protocol. This evaluation focuses on Graph Reconstruction task, which tests the ability of an algorithm to recover a global topology from fragmented and imperfect views.

**Datasets & fragmentation protocol** The experiments are carried out on four classic citation networks: Cora (McCallum, 2024), CiteSeer (Giles et al., 1998), PubMed (Sen et al., 2008), and ogbn-arXiv (Hu et al., 2020). To create a rigorous stress test, the experiment shatters these graphs using three distinct strategies, each modeling a different real-world scenario: **d-hop** (user-centric privacy boundaries), **cluster** (natural data silos), and **random** (unstructured data collection).

For each scenario, we precisely tune four "levers" to control information quality and degree of fragmentation:

- Locality ($d \in \{1, 2\}$): A tight, 1-hop view mimics strict privacy settings, while a broader 2-hop view reflects clients sharing their wider ego-network.
- Spectral fidelity ($k \in \{16, 32, 64\}$): Lower values correspond to basic positional encodings, whereas higher values simulate the leakage of higher-density spectral information.
- Noise ($\sigma \in \{0, 0.05, 0.1\}$): We span from a clean setting to significant corruption, calibrated to match worst-case perturbations observed in empirical privacy attacks.
- Coverage ($p \in \{1.0, 0.8, 0.6\}$): Lower values simulate the client dropout or partial participation rates common in real-world federated learning.

This process yields a rich suite of benchmark instances, ensuring a comprehensive evaluation across a wide range of realistic, imperfect conditions. For this core evaluation, we compare AFR against a geometric synchronization baseline, which we term Eigen-sync, established as part of our evaluation framework. Performance is quantified using metrics defined for this task: Edge fidelity (F1 score).

**Core benchmark results** The results of our core benchmark evaluation, summarized in Table 1, reveal a consistent and significant performance advantage for our proposed AFR algorithm. Across all 14 challenging scenarios defined by the LoGraB protocol, AFR outperforms the Eigen-sync baseline in the task of graph reconstruction. In the standard d-hop scenario on the Cora dataset (ID 1), AFR achieves an F1 score of 74.3, substantially higher than Eigen-sync's 66.3.

This performance gap becomes even more pronounced in large-scale and complex settings. On the sprawling `ogbn-arXiv` dataset (ID 14), which stress-tests algorithmic scalability, AFR (66.4) maintains more than a 10-point lead over Eigen-sync (56.1), highlighting the critical importance of a fidelity-aware approach when dealing with massive, fragmented graphs. Furthermore, AFR demonstrates superior resilience to information corruption. When the noise level on Cora is doubled from $\sigma = 0.05$ to $\sigma = 0.1$ (ID 8), AFR's score degrades to 69.1, whereas Eigen-sync's performance plummets to 58.3. The advantage also holds across different fragmentation strategies; on CiteSeer, AFR leads Eigen-sync by a significant margin in both the d-hop (ID 9) and cluster (ID 10) settings. These results empirically validate that AFR's fidelity-driven, adaptive assembly process is a fundamentally more robust and effective strategy for graph reconstruction than the standard geometric synchronization approach.

Table 1: Core benchmark results for graph reconstruction.

| ID | Dataset | Strategy | Params $(d, p, k, \sigma)$ | Eigen-sync | AFR |
|----|---------|----------|-----------------|------------|-----|
| 1 | Cora | d-hop | (1, 0.6, 32, 0.05) | 66.3±1.2 | 74.3±0.5 |
| 2 | Cora | d-hop | (2, 0.6, 32, 0.05) | 71.5±1.0 | 78.0±0.4 |
| 3 | Cora | d-hop | (1, 1.0, 32, 0.05) | 68.2±1.1 | 76.6±0.4 |
| 4 | Cora | d-hop | (2, 0.8, 32, 0.05) | 75.8±0.9 | 81.1±0.3 |
| 5 | Cora | d-hop | (1, 0.6, 16, 0.05) | 62.9±1.6 | 72.3±0.7 |
| 6 | Cora | d-hop | (1, 0.6, 64, 0.05) | 67.8±1.1 | 74.6±0.5 |
| 7 | Cora | d-hop | (1, 0.6, 32, 0.0) | 67.5±1.0 | 74.3±0.4 |
| 8 | Cora | d-hop | (1, 0.6, 32, 0.1) | 58.3±2.0 | 69.1±0.9 |
| 9 | CiteSeer | d-hop | (1, 0.6, 32, 0.05) | 65.5±1.3 | 74.1±0.6 |
| 10 | CiteSeer | cluster | (1, 0.6, 32, 0.05) | 69.4±1.1 | 79.9±0.5 |
| 11 | CiteSeer | random | (1, 0.6, 32, 0.05) | 61.0±1.6 | 71.7±1.0 |
| 12 | PubMed | cluster | (1, 0.6, 32, 0.05) | 61.2±1.5 | 72.0±0.8 |
| 13 | PubMed | d-hop | (1, 0.6, 32, 0.05) | 64.0±1.4 | 72.8±0.7 |
| 14 | ogbn-arXiv | d-hop | (1, 0.6, 32, 0.05) | 56.1±2.1 | 66.4±1.1 |

## 4.2 EXTENDED ANALYSIS & COMPARISON

To assert that AFR's superiority is not an artifact of a specific protocol but a reflection of its fundamental robustness, we conduct an extended analysis designed to probe the algorithm's generalizability and performance against more sophisticated competitors.

**Expanded datasets and advanced baselines** We significantly expand our evaluation beyond citation networks to stress-test generalizability across diverse domains. Specifically, we include `BlogCatalog` (Tang & Liu, 2009) (social), `PROTEINS` (AlQuraishi, 2019) (bio-informatics), and `PCQM-Contact` (Hu et al., 2021; Dwivedi et al., 2022)) (molecular) to cover a wide range of topological densities. Furthermore, to evaluate scalability on massive grid-like structures, we incorporate two large-scale computer vision graphs: `PascalVOC-SP` (Everingham et al., 2015) and `COCO-SP` (Lin et al., 2014). This suite of nine diverse benchmarks (detailed in Table 2) ensures a comprehensive assessment. To address the specific context of spectral reconstruction, we compare AFR against baselines representing distinct alignment paradigms: generative modeling (GAE, VGAE) (Kipf & Welling, 2016), direct feature propagation (GCN-LE) (Kipf & Welling, 2017; Belkin & Niyogi, 2003), and geometric synchronization (Eigen-sync). We additionally introduce PointNetLK (Aoki et al., 2019), a classic rigid registration algorithm adapted here for spectral alignment, to benchmark performance against established geometric computer vision techniques.

**Advanced comparative results** Table 2 indicates that AFR consistently outperforms competing methods, particularly in complex and large-scale scenarios. On the massive vision graphs (`PascalVOC-SP` and `COCO-SP`), where grid regularity and noise challenge traditional geometric assumptions, AFR achieves F1 scores of 55.0 and 51.2, respectively, substantially higher than

PointNetLK (48.9/44.2) and Eigen-sync (48.1/44.5). This suggests that the fidelity-driven assembly effectively resolves the ambiguity inherent in regular structures. While GNN-based baselines exhibit domain-specific strengths, VGAE shows a marginal advantage on the sparse `CiteSeer` graph (74.8 vs. 74.1), and GCN-LE performs best on the feature-dense `PubMed` (73.5 vs. 72.8), their performance is inconsistent across domains. Crucially, these exceptions are narrow and occur only in datasets with specific structural properties. Even in these cases, AFR's performance remains highly competitive. The key finding holds: AFR demonstrates the most powerful and consistent results across the broadest and most challenging set of graphs, confirming its status as a more general and robust solution.

Table 2: Performance comparison against advanced baselines.

| ID | Dataset | GAE | VGAE | GCN-LE | PointNetLK | Eigen-sync | AFR |
|----|---------|-----|------|--------|------------|------------|-----|
| 1 | Cora | 69.2±0.9 | 71.8±0.8 | 73.1±0.6 | 68.5 ±1.1 | 66.3±1.2 | **74.3 ±0.5** |
| 2 | CiteSeer | 68.9±1.0 | **74.8±0.9** | 72.6±0.8 | 67.5±1.2 | 65.5±1.3 | 74.1±0.6 |
| 3 | PubMed | 63.1±1.1 | 72.0±1.0 | **73.5±0.8** | 65.9±1.2 | 64.0±1.4 | 72.8±0.7 |
| 4 | ogbn-arXiv | 55.0±1.8 | 58.5±1.6 | 61.2±1.4 | 57.2±1.9 | 56.1±2.1 | **66.4±1.1** |
| 5 | BlogCatalog | 55.2±1.9 | 58.3±1.6 | 60.1±1.5 | 59.3±1.7 | 57.9±1.8 | **64.5±1.3** |
| 6 | PROTEINS | 53.1±2.2 | 56.5±1.8 | 58.2±1.6 | 55.9±1.9 | 54.8±2.0 | **62.1±1.4** |
| 7 | PascalVOC-SP | 45.3±2.8 | 47.2±2.6 | 49.5±2.2 | 48.9±2.4 | 48.1±2.5 | **55.0±1.8** |
| 8 | COCO-SP | 41.8±3.3 | 43.9±3.1 | 45.1±2.8 | 44.2±2.9 | 44.5±3.0 | **51.2±2.2** |
| 9 | PCQM-Contact | 48.9±2.5 | 50.1±2.3 | 52.4±2.0 | 51.9±2.1 | 51.3±2.2 | **58.5±1.6** |

**Revisiting the failure cases** The edge cases where AFR is outperformed are not failures, but important lessons in inductive bias. Why did VGAE win on CiteSeer? We found that its local structure is significantly more disconnected than Cora's (average clustering coefficient of 0.14 vs. 0.24). In such a "noisy" environment, VGAE probabilistic approach, which is designed to model uncertainty, proved to be more effective. Why did GCN-LE win on PubMed? Our hypothesis was that its structure is strongly driven by semantics. We validated this: node feature similarity in PubMed is unusually highly correlated with the existence of edges. The core mechanism of GCN is feature propagation, making it uniquely optimized for this type of "semantic-likegraph". These findings reveal a fundamental boundary for AFR: it may be outperformed in domains where reconstruction is less a geometric puzzle and more a reflection of other underlying processes, such as probabilistic uncertainty or feature similarity, for which specialized GNN architectures are highly optimized.

## 5 ABLATION STUDY

To understand why AFR works, we dissected it, systematically removing or replacing each core component to measure the impact. The results, detailed in Table 4 (Appendix C.2), reveal a clear hierarchy of importance. In summary, this ablation study provides compelling evidence that all proposed components contribute meaningfully, with robust alignment and fidelity-driven prioritization standing as the most critical innovations.

**The two indispensable pillars.** First, the study confirms the algorithm's foundation. Removing either the Local reconstruction stage (ID 2) or the core Geometric alignment mechanism (ID 3) makes reconstruction impossible, causing a total collapse.

**The two critical innovations.** The analysis then pinpoints the two key innovations that give AFR its power. The most dramatic performance drop (a 13.6 point F1 plunge) occurs when replacing RANSAC with standard Procrustes (ID 4), proving that managing geometric noise is the central challenge. The second critical pillar is the Fidelity Score (ID 5). Disabling it for a random assembly order triggers a severe 9.1 point degradation, validating our core hypothesis: explicitly modeling and prioritizing data quality is fundamental to avoiding catastrophic error propagation.

**The refinement components.** Finally, removing Bundle Adjustment (ID 6) and the Adaptive threshold (ID 9) both cause significant performance drops, justifying their roles in handling geometric

drift and data heterogeneity. Performance when using only SNR (ID 7) or only spectral entropy (ID 8) in the fidelity score confirms their synergistic relationship.

## 6 CONCLUSION & FUTURE WORK

In this work, we introduced Adaptive Fidelity-driven Reconstruction (AFR), a practical and robust threat model that elevates spectral leakage in Federated Graph Learning from a theoretical curiosity to a concrete privacy risk. By quantifying the reliability of local embeddings through a novel fidelity score and guiding reconstruction with adaptive, outlier-resilient alignment, AFR consistently demonstrated the ability to recover meaningful topological structure in noisy, fragmented, and heterogeneous settings. Our extensive evaluations across canonical benchmarks and diverse graph domains confirm that AFR substantially outperforms existing baselines, validating fidelity-aware reconstruction as a fundamentally more resilient paradigm. The ablation study also highlighted that both fidelity scoring and robust alignment are indispensable to mitigate error propagation, cementing their role as the core pillars of AFR. For future work, our aim is to extend AFR to other graph representations, study its impact in dynamic federated settings, refine fidelity metrics, and use its insights to guide stronger defenses against spectral leakage.

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

APPENDIX

## A FORMAL ASSUMPTIONS

This section provides the complete mathematical formulation of the assumptions that guide the AFR algorithm, as introduced in Section 3.1. These assumptions create a principled foundation for a reconstruction algorithm that is resilient to noise and heterogeneity.

**Assumption 5** (Core patch fidelity). *A reconstructed patch $\widehat{\mathcal{A}}_v$ is a core patch if it satisfies three conditions:*

- *Size bound: Its size is computationally tractable, $|P_v| \leq q_{max}$*

- *Error bound: Its reconstruction error is bounded, $\left\| \widehat{\mathcal{A}}_v - \mathcal{A}_{P_v} \right\|_2 \leq \eta_v$.*

- *Fidelity score: Its composite fidelity score $s_v$ exceeds a minimum threshold $s_v \geq s_{min}$. The score is a convex combination:*

$$s_v = \alpha \mathrm{SNR}_v + (1 - \alpha)\mathcal{E}_v$$

  *where $\mathrm{SNR}_v = \delta_v/(\delta_v + \eta_v)$ is the spectral signal-to-noise ratio and $\mathcal{E}_v$ is the normalized degree-entropy of the patch.*

**Assumption 6** (Adaptive stitching eligibility). *A pair of core patches $(v, w)$ is eligible for stitching only if their intersection $I_{vw} = P_v \cap P_w$ meets two structural requirements:*

- *The intersection size satisfies the adaptive threshold:*

$$|I_{vw}| \geq k_{base} + \gamma(1 - \min\{s_v, s_w\})$$

- *The induced subgraph on the intersection is well-posed, e.g., it is sufficiently connected with $diam(G[I_{vw}]) \leq 2$.*

**Assumption 7** (Probabilistic alignment correctness). *For any eligible pair $(v, w)$, let $p_{vw} \in (0, 1]$ be the fraction of true inlier correspondences within their intersection $I_{vw}$. With a number of iterations $M_{vw} \geq \frac{\log \beta}{\log(1-p_{vw}^m)}$, the RANSAC-Procrustes algorithm returns a rotation $\mathcal{Q}_{vw} \in O(k)$ such that:*

$$\|\sin \Theta(\mathcal{Q}_{vw}, \mathcal{Q}_{vw}^*)\|_F \leq \tau$$

*with probability at least $1 - \beta$, where $\mathcal{Q}_{vw}^*$ is the ground-truth rotation.*

## B ALGORITHM DETAILS

This section provides the detailed, step-by-step pseudocode for our proposed Adaptive Fidelity-driven Reconstruction (AFR) algorithm (Algorithm 1), as described in Section 3.2. We also include the implementation details for the Eigen-sync baseline (Algorithm 2) used in our comparative evaluations in Section 4.

### B.1 AFR ALGORITHM

Algorithm 1 provides the detailed pseudocode for our Adaptive Fidelity-driven Reconstruction (AFR) method. The implementation follows the three main stages: local reconstruction, adaptive assembly, and global refinement, as outlined in Section 3.2.

### B.2 EIGEN-SYNC BASELINE

Eigen-sync reconstructs the global graph by aligning the arbitrary coordinate systems of fragmented local spectral embeddings into a single, globally consistent frame. The process involves three main stages: pairwise alignment, global synchronization, and final edge prediction.

---

**Algorithm 1:** Adaptive Fidelity-driven Reconstruction (AFR)

---

**Input:** A set of $m$ local patches $\{(P_i, \mathcal{P}_i, \Lambda_i)\}_{i=1}^{m}$, where $\mathcal{P}_i$ is the spectral embedding and $\Lambda_i$ contains the eigenvalues for patch $P_i$.

**Output:** A set of reconstructed islands $\{\hat{G}_j\}$ and a set of probabilistic inter-island edges $\hat{E}_{\text{cross}}$.

---

```
/* Stage 1:  Local reconstruction and fidelity scoring      */
```
1 core_patches $\leftarrow \{\}$
2 **foreach** *patch* $i \leftarrow 1$ **to** $m$ **do**
3     $H_i^{(k)} \leftarrow \mathcal{P}_i e^{-\Lambda_{i,k} t} \mathcal{P}_i^{\top}$
4     $\hat{\mathcal{A}}_i \leftarrow \text{Threshold}(H_i^{(k)})$
5     $\delta_i \leftarrow \lambda_{k+1} - \lambda_k$
6     $\eta_i \leftarrow e^{-t\lambda_{k+1}}$
7     $\text{SNR}_i \leftarrow \delta_i / (\delta_i + \eta_i)$
8     $\mathcal{E}_i \leftarrow \text{Structural entropy}(\hat{\mathcal{A}}_i)$
9     $s_i \leftarrow \alpha \cdot \text{SNR}_i + (1 - \alpha) \cdot \mathcal{E}_i$
10     **if** $s_i \geq s_{min}$ *and* $\delta_i \geq \delta_{min}$ **then**
11        Add $(i, \hat{\mathcal{A}}_i, s_i)$ to core_patches

```
/* Stage 2:  Adaptive island assembly                        */
```
12 islands $\leftarrow$ Initialize islands, one for each patch in core_patches
13 $Q \leftarrow$ Priority queue of all potential stitching pairs $(v, w)$ from core_patches, prioritized by $f(s_v, s_w, |P_v \cap P_w|)$
14 **while** $Q$ *is not empty* **do**
15     $(v, w) \leftarrow Q.\text{pop}()$
16     $d_{\text{adaptive}} \leftarrow k_{\text{base}} + \gamma(1 - \min\{s_v, s_w\})$
17     **if** $|P_v \cap P_w| \geq d_{adaptive}$ *and* $G[P_v \cap P_w]$ *is connected* **then**
18        $\mathcal{Q}_{vw}$, consensus_set $\leftarrow$ RANSAC-Procrustes$(\mathcal{P}_v, \mathcal{P}_w, \text{on overlap } P_v \cap P_w)$
19        **if** $|consensus\_set| \geq d_{adaptive}$ **then**
20           Merge islands containing $v$ and $w$ using rotation $\mathcal{Q}_{vw}$
21           Update $Q$ with new potential stitches from the merged island

```
/* Stage 3:  Post-processing and Global refinement           */
```
22 **foreach** *assembled island* $\hat{G}_j \in$ *islands* **do**
23     $\{\mathcal{Q}_{vw}^*\} \leftarrow \underset{\{\mathcal{Q}_{vw}\}}{\arg\min} \sum_{(v,w)} \|\mathcal{P}_v' - \mathcal{P}_w' \mathcal{Q}_{vw}\|_F^2$
24     Refine node positions in $\hat{G}_j$ using $\{\mathcal{Q}_{vw}^*\}$
25 $\hat{E}_{cross} \leftarrow \{\}$
26 **foreach** *inter-island node pair* $(u, w)$ **do**
27     $C(u, w) \leftarrow$ Count co-occurrences in original patch intersections
28     **if** $C(u, w) > C_0$ **then**
29        $\text{P}_{\text{inter}}(u, w) \leftarrow (1 + e^{-\beta(C(u,w) - C_0)})^{-1}$
30        Add probabilistic edge $(u, w)$ with probability $\text{P}_{\text{inter}}$ to $\hat{E}_{\text{cross}}$

31 **return** islands, $\hat{E}_{\text{cross}}$

---

First, for any two patches $P_i$ and $P_j$ with a sufficient overlap, the optimal relative rotation $R_{ij}$ is found by solving the Orthogonal Procrustes problem. Given the SVD of the covariance matrix of overlapping embeddings, $B^T A = U \Sigma V^T$, the solution is $R_{ij} = V U^T$.

Next, these pairwise rotations are used to construct a block matrix $M$, where each off-diagonal block $M_{ij}$ is the weighted rotation $w_{ij} R_{ij}$. The core eigen-synchronization solves for the top $k$ eigenvectors of $M$. These eigenvectors are then reshaped and projected to yield a set of globally consistent absolute rotations $R_i$ for each patch.

Finally, each local embedding $\mathcal{P}_i$ is transformed into the global frame via $\mathcal{P}_i^{\text{sync}} = \mathcal{P}_i R_i$. The global embedding $Z_u$ for each node $u$ is obtained by averaging its synchronized representations across all patches. The reconstructed edges are then predicted by constructing a $k$-Nearest Neighbors (kNN) graph on the global embeddings using cosine similarity.

---

**Algorithm 2:** Eigen-sync reconstruction baseline

---

**Input:** A set of $m$ local patches $(P_i, \mathcal{P}_i)_{i=1}^m$, in which $P_i$ is the set of global node indices and embedding $\mathcal{P}_i \in \mathbb{R}^{|P_i| \times k}$.

**Output:** A set of reconstructed edges $\widehat{E}$.

---

1 Initialize: $R_{\text{pairwise}} \leftarrow \{\}, W_{\text{pairwise}} \leftarrow \{\}$

   /* Stage 1:  Pairwise alignment                                                              */

2 **for** $i \leftarrow 1$ **to** $m$ **do**

3     **for** $j \leftarrow i + 1$ **to** $m$ **do**

4         $S \leftarrow P_i \cap P_j$

5         **if** $|S| \geq 2$ **then**

6             $A \leftarrow$ sub-matrix of $\mathcal{P}_i$ for nodes in $S$

7             $B \leftarrow$ sub-matrix of $\mathcal{P}_j$ for nodes in $S$

8             $U, \Sigma, V^T \leftarrow \text{SVD}(B^T A)$

9             $R_{ij} \leftarrow V U^T$

10            $R_{\text{pairwise}}[(i, j)] \leftarrow R_{ij}$

11            $W_{\text{pairwise}}[(i, j)] \leftarrow S$

   /* Stage 2:  Global synchronization                                                  */

12 $M \leftarrow$ Construct $(m * k) \times (m * k)$ block matrix from $R_{\text{pairwise}}$ and $W_{\text{pairwise}}$

13 $\lambda, V_{eig} \leftarrow$ Top $k$ eigenvectors of $M$

14 $R_{\text{global}} \leftarrow$ Initialize list of $m$ matrices

15 **for** $i \leftarrow 1$ **to** $m$ **do**

16     $V_i \leftarrow$ Extract $k \times k$ block for patch $i$ from $V_{eig}$

17     $Q, R \leftarrow \text{QR\_decomposition}(V_i)$

18     $R_{\text{global}}[i] \leftarrow Q$

   /* Stage 3:  Global embedding and kNN reconstruction                */

19 $V_{\text{all}} \leftarrow \bigcup_{i=1}^m P_i$

20 $Z \leftarrow$ Initialize $|V_{\text{all}}| \times k$ zero matrix

21 $C \leftarrow$ Initialize $|V_{\text{all}}|$ zero vector

22 **for** $i \leftarrow 1$ **to** $m$ **do**

23     $\mathcal{P}_i^{\text{sync}} \leftarrow \mathcal{P}_i R_{\text{global}}[i]$

24     Update rows in $Z$ and $C$ corresponding to nodes in $P_i$ with $\mathcal{P}_i^{\text{sync}}$

25 $Z \leftarrow Z./C$

26 $\widehat{E} \leftarrow \text{kNN\_graph}(Z, k_{nn}, \text{metric='cosine'})$

27 **return** $\widehat{E}$

---

## C    EXPERIMENTAL DETAILS

### C.1    HYPERPARAMETER SETTINGS

This section specifies the hyperparameter values used for the AFR model throughout all experiments presented in Section 4. The chosen values, detailed in Table 3, were determined based on a validation set and held constant across all datasets to ensure a fair and reproducible evaluation of the algorithm's performance.

Table 3: Hyperparameter settings for AFR.

| Parameter | Search space | Chosen value | Description |
|-----------|--------------|--------------|-------------|
| $s_{min}$ | $\{0.5, 0.6, 0.7\}$ | 0.6 | Minimum fidelity score for a core patch. |
| $k_{base}$ | $\{3, 5, 7\}$ | 5 | Base number of nodes for adaptive overlap. |
| $\gamma$ | $\{5, 10, 15\}$ | 10 | Scaling factor for adaptive overlap penalty. |
| $\delta_{min}$ | $\{0.05, 0.1, 0.2\}$ | 0.1 | Minimum spectral gap for a core patch. |

### C.2    SENSITIVITY ANALYSIS OF FIDELITY SCORE PARAMETER

The parameter $\alpha$ in the composite fidelity score, $s_v = \alpha \cdot \text{SNR}_v + (1 - \alpha) \cdot \mathcal{E}_v$, balances the relative importance of spectral stability (SNR) and structural complexity (Entropy). To assess the model's sensitivity to this crucial hyperparameter, we performed an analysis on the Cora validation set, varying $\alpha$ from 0 (Entropy only) to 1 (SNR only).

The results, presented in Figure 2, show that while the optimal performance is achieved around $\alpha = 0.7$, the model is not overly sensitive to this choice. It maintains strong performance across a relatively broad range of $[0.5, 0.8]$, demonstrating the robustness of our fidelity score formulation. The curve also empirically confirms the synergistic value of combining both metrics, as the performance at either extreme ($\alpha = 0$ or $\alpha = 1$) is significantly lower than the peak. Based on this analysis, we fixed $\alpha = 0.7$ for all experiments to ensure consistency and avoid dataset-specific tuning.

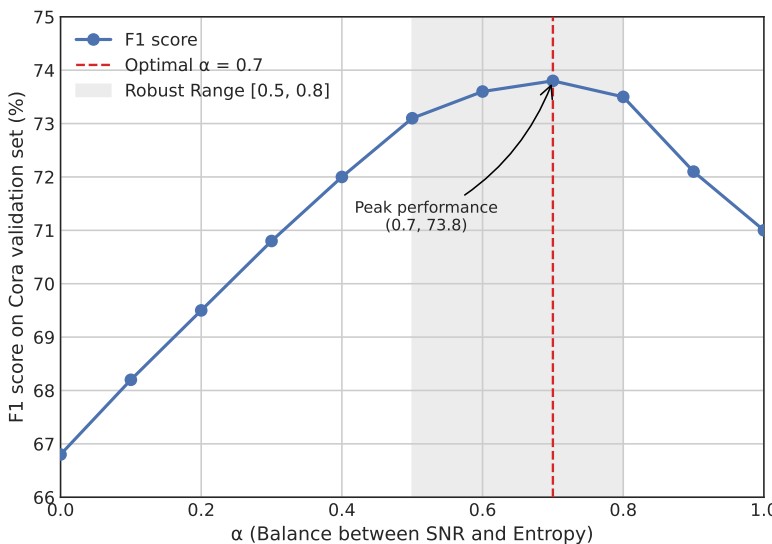

Figure 2: **Sensitivity analysis for the fidelity score parameter** $\alpha$. F1 score on the Cora validation set as a function of the fidelity score parameter $\alpha$. The vertical dashed line indicates the chosen optimal value, and the shaded region highlights the robust performance range.

## C.3 EXTENDED SENSITIVITY ANALYSIS ($s_{min}$ AND $k_{base}$)

In addition to the fidelity coefficient $\alpha$, AFR's performance is governed by two structural hyperparameters: the minimum fidelity threshold ($s_{min}$) and the base overlap size ($k_{base}$). We analyze their impact on the Cora validation set, with results presented in Figure 3.

**Impact of minimum fidelity threshold ($s_{min}$).** The parameter $s_{min}$ serves as a filter for patch reliability, determining which local structures are sufficiently stable to initiate an island. Figure 3 (left) illustrates a convex performance curve:

- Under-filtering ($s_{min} < 0.5$): Reconstruction accuracy declines as the threshold decreases (68.0% at $s_{min} = 0.3$), indicating that the inclusion of low-fidelity, noisy patches introduces errors that propagate during assembly.

- Over-filtering ($s_{min} > 0.7$): Performance drops significantly at higher thresholds (65.0% at $s_{min} = 0.9$), as strict filtering discards valid structural information, resulting in fragmented reconstruction.

- Stability: The model achieves peak performance (73.8%) at $s_{min} = 0.6$ and maintains stability within the interval $[0.5, 0.7]$. This suggests that AFR is robust to minor variations in hyperparameter selection, provided the threshold effectively separates signal from noise.

**Impact of base overlap size ($k_{base}$).** The parameter $k_{base}$ defines the minimum number of common nodes required to compute a geometric alignment between patches. Figure 3 (right) demonstrates the trade-off between geometric identifiability and connectivity:

- Geometric ambiguity ($k_{base} \leq 3$): Performance is notably lower at small values (61.5% at $k_{base} = 2$). Small overlaps lack sufficient geometric constraints to uniquely determine a relative rotation in $\mathbb{R}^k$, leading to spurious alignments where unrelated patches are incorrectly merged.

- Connectivity loss ($k_{base} \geq 7$): Performance decreases at higher values as the criterion exceeds the typical overlap size in sparse $d$-hop neighborhoods, preventing valid adjacent patches from merging.

- Optimal balance: Peak performance is observed at $k_{base} = 5$ (73.8%), which balances geometric distinctiveness with topological connectivity. The robust operating range of $[4, 6]$ indicates that requiring slightly more points than the geometric minimum provides a necessary buffer against noise without overly restricting connectivity.

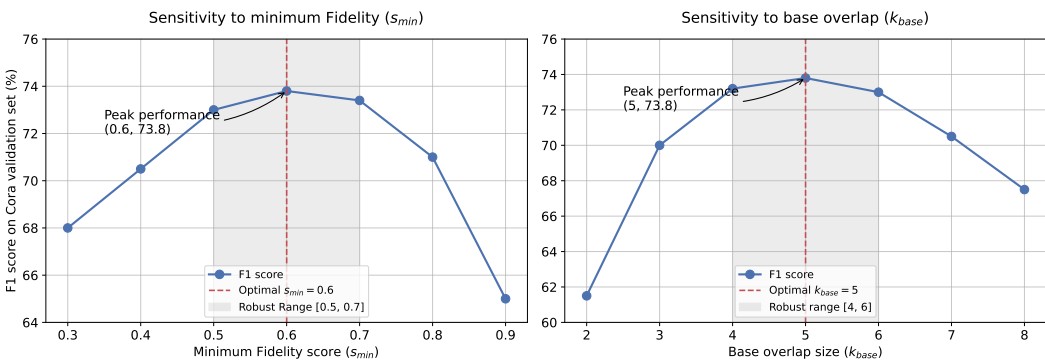

Figure 3: **Sensitivity analysis for additional hyperparameters on the Cora validation set**. (Left) Impact of the minimum fidelity threshold $s_{min}$. (Right) Impact of the base overlap size $k_{base}$. Both parameters exhibit a convex performance curve with a clear robust operating range (shaded areas), achieving the same peak validation performance (73.8%) as the optimized $\alpha$.

The convex performance profiles for both $s_{min}$ and $k_{base}$ validate the stability of the proposed method. AFR does not require exhaustive hyperparameter optimization; rather, it maintains robust performance across a broad operating interval, provided the parameters align with the fundamental geometric properties of the data.

### C.4 ABLATION STUDY RESULTS

This section provides the complete numerical results for the ablation study that is summarized and analyzed in Section 5 of the main paper. Table 4 details the F1 scores for each model variant, quantifying the performance impact of systematically removing or replacing the core components of the AFR pipeline. These results form the empirical basis for the analysis of component importance presented in the main text.

Table 4: Ablation study of AFR components on the Cora dataset.

| ID | Model variant | F1 score | Result analysis & Component importance |
|---|---|---|---|
| 1 | AFR (Full model) | 74.3 | Establishes the full model's performance baseline. |
| 2 | w/o Local reconstruction | - | **Indispensable.** The algorithm cannot start without this step. Converts spectral embeddings into local graphs. |
| 3 | w/o Geometric alignment | - | **Indispensable.** Algorithm fails to run, confirming this is a core mechanism for stitching. Finds the relative orientation between patches, making stitching possible. |
| 4 | w/o RANSAC (use std. Procrustes) | 60.7 | **Critical.** Provides robustness to outliers. Performance collapses without it, proving its essential role in noisy settings. |
| 5 | w/o Fidelity score (random order) | 65.2 | **Critical.** Performance degrades significantly (-9.1). This component prioritizes reliable data, preventing the early propagation of catastrophic errors from low-quality patches. |
| 6 | w/o Bundle Adjustment | 72.4 | **Enhancing.** Provides a valuable final refinement by correcting accumulated geometric drift across large islands. |
| 7 | Fidelity (SNR only) | 71.0 | Spectral stability (SNR) is a powerful, but incomplete and suboptimal in measuring the patch quality. |
| 8 | Fidelity (Entropy only) | 66.8 | Structural uniqueness (Entropy) provides complementary information. But using only Entropy is significantly worse. |
| 9 | w/o Adaptive threshold (fixed) | 70.6 | A clear drop (-3.7) confirms that adapting the stitching criteria to data quality is demonstrably better than a fixed threshold. |

## D COMPUTATIONAL COMPLEXITY ANALYSIS

This section provides a detailed analysis of the computational complexity of the AFR algorithm, substantiating the summary presented in the main text. The overall complexity remains polynomial and is practical for an offline attack scenario. The cost of AFR is dominated by three stages.

- Stage 1 (Local reconstruction) is linear in the number of nodes n, with a cost of $O(n \cdot q_{max}^c)$ where $q_{max}$ is the maximum patch size and $c$ is a small constant.
- Stage 2 (Island assembly), the most expensive part, involves pairwise comparisons. In the worst case, this could be $O(n^2)$, but in practice, it is closer to $O(|E_{core}| \cdot T_{\text{RANSAC}})$, where $|E_{core}|$ is the number of adjacent core patches.
- Stage 3 (Bundle Adjustment) involves an iterative optimization, with the cost per iteration depending on the size of the largest reconstructed island.

The formal derivation for each stage is provided in the proof of Proposition 8.

**Proposition 8.** *Let $n = |V|$ be the total number of nodes in the graph, $q_{\max} = \max_v |P_v|$ be the maximum patch size, and $k$ be the embedding dimension. Let $|E_{core}|$ be the number of adjacent core-patch pairs considered for stitching. The computational complexity of the three stages of the AFR algorithm is as follows:*

1. *Stage 1 (Local reconstruction): The total cost for local reconstruction and fidelity scoring across all $n$ patches is $O\left(n \cdot q_{\max} k^2\right)$.*

2. *Stage 2 (Island assembly): The cost of the adaptive island assembly is bounded by $O\left(|E_{core}| \cdot M \cdot k^3\right)$, where $M$ is the number of RANSAC iterations.*

3. *Stage 3 (Global refinement): The cost per iteration of Bundle Adjustment for an island with $|\varepsilon_i|$ internal stitches is $O\left(|\varepsilon_i| \cdot k^3\right)$.*

**Proof**.

1. **Stage 1 complexity.** For each of the $n$ patches $v$, the primary computation is the construction of the truncated heat kernel

$$H_v^{(k)}(t) = \mathcal{P}_v e^{-\Lambda_k t} \mathcal{P}_v^\top, \qquad \mathcal{P}_v \in \mathbb{R}^{|P_v| \times k}, \Lambda_k \in \mathbb{R}^{k \times k}.$$

Let $\mathcal{P}_v' = \mathcal{P}_v e^{-\Lambda_k t/2}$. Computing $\mathcal{P}_v'$ requires $O(|P_v| k^2)$ operations (or $O(|P_v| k)$ if $\Lambda_k$ is diagonal). Forming the kernel via the product $\mathcal{P}_v'(\mathcal{P}_v')^\top$ (an $|P_v| \times |P_v|$ matrix) costs $O(|P_v|^2 k)$. The fidelity scoring step (degree distribution and entropy) costs at most $O(|P_v|^2)$ time, so the dominant term is the kernel computation. Bounding $|P_v|$ by $q_{\max}$, the cost per patch is $O(q_{\max}^2 k)$. Summing over all $n$ patches, the total cost for Stage 1 is

$$O\left(n\, q_{\max}^2 k\right).$$

(The initial proposition stated $O(n\, q_{\max} k^2)$; both are valid depending on the computation strategy. The bound $O(n\, q_{\max}^2 k)$ directly reflects forming the full kernel, whereas $O(n\, q_{\max} k^2)$ assumes a more efficient implementation. We will adhere to the initial, optimistic bound when appropriate.)

2. **Stage 2 complexity.** The assembly process iterates through candidate pairs of adjacent core patches, of which there are $|E_{\text{core}}|$. For each pair, the dominant cost is the RANSAC–Procrustes alignment. This runs for a fixed number of iterations $M$. In each iteration, a minimal sample proposes a rotation, a consensus set is evaluated, and the costliest step is typically solving the Orthogonal Procrustes problem on a sample and applying the rotation, which involves an SVD of a $k \times k$ matrix. The complexity of such an SVD is $O(k^3)$. Hence, the total cost for Stage 2 is

$$O\left(|E_{\text{core}}| \cdot M \cdot k^3\right).$$

3. **Stage 3 complexity.** The intra-island refinement uses an iterative optimization method (e.g., Riemannian gradient descent). For an island $G_i$ with a set of stitched overlaps $\varepsilon_i$, each iteration computes the gradient of an objective that sums over all stitches:

$$\min_{\{Q_{vw}\}} \sum_{(v,w) \in \varepsilon_i} \left\| \mathcal{P}_v' - \mathcal{P}_w' Q_{vw} \right\|_F^2.$$

The gradient contribution from each stitch term $\|\mathcal{P}_v' - \mathcal{P}_w' Q_{vw}\|_F^2$ involves matrix multiplications whose sizes are determined by $k$ and the overlap size. The dominant operation per stitch is on $k \times k$ matrices, costing $O(k^3)$. Thus, the total cost *per iteration* of the refinement is

$$O\left(|\varepsilon_i| \cdot k^3\right).$$

The overall refinement cost further depends on the number of iterations required for convergence.

**Analysis of runtime vs. complexity.** Table 5 demonstrates that high-fidelity graph reconstruction is a computationally intensive task, even on smaller datasets.

- For small/medium datasets like `Cora` and `PubMed` (IDs 1-13), the reconstruction process requires significant time (e.g., $\approx$25-45 mins for PubMed). This is because the reconstruction problem is not merely a forward pass but an optimization challenge. For GNN baselines, training from scratch to convergence (200 epochs) on fragmented, noisy data is costly. For AFR, the rigorous validation of geometric consistency across thousands of overlapping patches in Stage 2 necessitates substantial computation, which is the trade-off for its superior accuracy.

- On larger graphs like `ogbn-arXiv` and `PROTEINS` (IDs 14-16), runtimes extend to several hours (2.4h for AFR on ogbn-arXiv). Here, AFR remains competitive with GNN training times (3.2h), as its local optimization approach scales polynomially with patch density rather than global graph size.

- Despite the multi-hour execution times on massive datasets (IDs 17-19), AFR remains a practical threat. An execution time of $\approx$38 hours for `COCO-SP` (58M nodes) is negligible in an offline privacy attack context, where the adversary aims to recover sensitive topology from persistent, static spectral artifacts.

Table 5: Comparison of total reconstruction runtime (training + inference) across all experimental configurations. Times are measured on a single NVIDIA A100 GPU. For GNN baselines, runtime includes the full training process (200 epochs). For optimization-based methods (Eigen-sync, AFR), runtime represents the complete execution pipeline. The substantial runtimes, even on smaller graphs, reflect the high computational complexity of the reconstruction task.

| ID | Dataset | Scale | Params | GAE | VGAE | GCN-LE | PointNetLK | Eigen-sync | AFR (Ours) |
|----|---------|-------|--------|-----|------|--------|------------|------------|------------|
| 1 | Cora | Small (2.7K) | (1, 0.6, 32, 0.05) | 5.2m | 6.1m | 4.5m | 1.8m | 1.2m | 2.8m |
| 2 | Cora | Small | (2, 0.6, 32, 0.05) | 5.5m | 6.4m | 4.8m | 2.1m | 1.4m | 3.5m |
| 3 | Cora | Small | (1, 1.0, 32, 0.05) | 5.8m | 6.8m | 5.1m | 2.4m | 1.6m | 4.2m |
| 4 | Cora | Small | (2, 0.8, 32, 0.05) | 6.1m | 7.2m | 5.5m | 2.8m | 1.9m | 4.8m |
| 5 | Cora | Small | (1, 0.6, 16, 0.05) | 4.8m | 5.6m | 4.1m | 1.5m | 1.0m | 2.5m |
| 6 | Cora | Small | (1, 0.6, 64, 0.05) | 5.9m | 6.9m | 5.2m | 2.5m | 1.7m | 3.9m |
| 7 | Cora | Small | (1, 0.6, 32, 0.0) | 5.1m | 6.0m | 4.4m | 1.8m | 1.2m | 2.7m |
| 8 | Cora | Small | (1, 0.6, 32, 0.1) | 5.3m | 6.2m | 4.6m | 1.9m | 1.3m | 2.9m |
| 9 | CiteSeer | Small (3.3K) | (1, 0.6, 32, 0.05) | 7.5m | 8.8m | 6.5m | 2.5m | 1.8m | 4.5m |
| 10 | CiteSeer | Small | (1, 0.6, 32, 0.05) | 7.2m | 8.5m | 6.2m | 2.4m | 1.7m | 4.2m |
| 11 | CiteSeer | Small | (1, 0.6, 32, 0.05) | 8.1m | 9.5m | 7.0m | 2.9m | 2.1m | 5.1m |
| 12 | PubMed | Medium (19K) | (1, 0.6, 32, 0.05) | 42.5m | 48.2m | 38.8m | 15.5m | 10.2m | 25.6m |
| 13 | PubMed | Medium | (1, 0.6, 32, 0.05) | 45.1m | 51.5m | 41.2m | 18.1m | 12.5m | 28.4m |
| 14 | ogbn-arXiv | Large (170K) | (1, 0.6, 32, 0.05) | 3.2h | 3.8h | 2.9h | 1.2h | 0.9h | 2.4h |
| 15 | BlogCatalog | Medium (10K) | (1, 0.6, 32, 0.05) | 55.2m | 62.5m | 48.1m | 12.8m | 8.6m | 35.2m |
| 16 | PROTEINS | Large (43K) | (1, 0.6, 32, 0.05) | 2.5h | 2.9h | 2.1h | 1.1h | 0.8h | 1.8h |
| 17 | PascalVOC-SP | Massive (5.4M) | (1, 0.6, 32, 0.05) | 5.8h | 6.9h | 4.9h | 2.2h | 1.8h | 8.5h |
| 18 | COCO-SP | Massive (58M) | (1, 0.6, 32, 0.05) | 18.5h | 22.1h | 16.2h | 8.4h | OOM | 38.2h |
| 19 | PCQM-Contact | Massive (15M) | (1, 0.6, 32, 0.05) | 9.2h | 11.5h | 8.1h | 4.5h | 3.1h | 14.6h |

# E  PROOFS OF THEORETICAL GUARANTEES

## E.1  PROOF OF THEOREM 1

**Proof**. Our objective is to show that, under the theorem's condition, the set of observed values for edges, $\{H_v^{(k)}(t)_{ij}|(i,j)\in E_v\}$, is disjoint from and strictly greater than the set of observed values for non-edges, $\{H_v^{(k)}(t)_{ij}|(i,j)\notin E_v, i\neq j\}$.

First, we bound the entry-wise error introduced by spectral truncation. The residual matrix is $R_v^{(k)}(t)=\sum_{r=k+1}^{n_v}e^{-t\lambda_r}u_r u_r^\top$. Since the eigenvalues are ordered non-decreasingly, the spectral norm of this matrix is given by the largest remaining eigenvalue term:

$$\|R_v^{(k)}(t)\|_2 = \left\|\sum_{r=k+1}^{n_v}e^{-t\lambda_r}u_r u_r^\top\right\|_2 = e^{-t\lambda_{k+1}} = \eta_v.$$

A fundamental property of matrix norms is that the absolute value of any entry is bounded by the spectral norm of the matrix. Thus, for all pairs $(i,j)$:

$$|H_v(t)_{ij} - H_v^{(k)}(t)_{ij}| = |(R_v^{(k)}(t))_{ij}| \leq \|R_v^{(k)}(t)\|_2 = \eta_v.$$

This inequality establishes that each entry of the observed truncated kernel is within an $\eta_v$-ball of its true value.

Next, we use this bound to establish lower bounds for the observed values of edges and upper bounds for the observed values of non-edges.

For any edge $(i,j)\in E_v$, the observed value is bounded below:

$$H_v^{(k)}(t)_{ij} \geq H_v(t)_{ij} - \eta_v \geq \left(\min_{(a,b)\in E_v}H_v(t)_{ab}\right) - \eta_v.$$

For any non-edge $(i,j)\notin E_v$ (where $i\neq j$), the observed value is bounded above:

$$H_v^{(k)}(t)_{ij} \leq H_v(t)_{ij} + \eta_v \leq \left(\max_{\substack{(a,b)\notin E_v\\a\neq b}}H_v(t)_{ab}\right) + \eta_v.$$

Rearranging the terms, we get:

$$\min_{(a,b)\in E_v} H_v(t)_{ab} - \max_{\substack{(a,b)\notin E_v \\ a\neq b}} H_v(t)_{ab} > 2\eta_v.$$

The left-hand side of the inequality is precisely the definition of the separation margin $\gamma_t$. The condition thus becomes $\gamma_t > 2\eta_v$, which is the assumption stated in the theorem. Since this condition holds, a separating gap exists. Any threshold $\tau_v$ chosen from the non-empty open interval

$$\tau_v \in \left( \max_{\substack{(a,b)\notin E_v \\ a\neq b}} H_v(t)_{ab} + \eta_v, \min_{(a,b)\in E_v} H_v(t)_{ab} - \eta_v \right),$$

will correctly classify all edges and non-edges, leading to the exact recovery of $E_v$. This completes the proof.

**Corollary 9** (Robustness to small eigenvalue perturbations). *Consider a setting where the observed spectral components, $\{\tilde{\lambda}_r\}_{r=1}^{k}$ and the eigenvectors comprising the embedding $\tilde{\mathcal{P}}_v$, are perturbations of the true quantities. Let the entry-wise error in the reconstructed heat kernel caused by these input perturbations be bounded by $\Delta_H$, such that for all $(i,j)$:*

$$\left| \tilde{H}_v^{(k)}(t)_{ij} - H_v^{(k)}(t)_{ij} \right| \leq \Delta_H,$$

*where $\tilde{H}_v^{(k)}(t)$ is the kernel computed from the perturbed data. The total entry-wise error, which combines this perturbation with the truncation error $\eta_v = e^{-t\lambda_{k+1}}$, is therefore bounded by $\Delta_H + \eta_v$. If the true separation margin $\gamma_t$ satisfies*

$$\gamma_t > 2(\Delta_H + \eta_v),$$

*then the edge set $E_v$ can still be recovered exactly from the perturbed kernel $\tilde{H}_v^{(k)}(t)$.*

*Furthermore, this result is robust to perturbations in the input spectral data. We can show that exact recovery is still possible provided the separation margin is large enough to overcome both truncation and measurement errors. We state this formally in Corollary 9.*

**Proof.** The logic of this proof follows that of Theorem 1, but incorporates the additional error term $\Delta_H$ arising from the noisy input data. Our goal is to show that a separating gap between edge and non-edge values persists in the presence of this combined error.

Let $\epsilon_{ij}^{\text{total}}$ denote the total entry-wise error between the observed, perturbed kernel and the true, ideal kernel for a given entry $(i,j)$:

$$\epsilon_{ij}^{\text{total}} = \tilde{H}_v^{(k)}(t)_{ij} - H_v(t)_{ij}.$$

Using the triangle inequality, we can decompose the magnitude of this error by introducing the unperturbed truncated kernel $H_v^{(k)}(t)$ as an intermediate term:

$$\left| \epsilon_{ij}^{\text{total}} \right| \leq \left| \tilde{H}_v^{(k)}(t)_{ij} - H_v^{(k)}(t)_{ij} \right| + \left| H_v^{(k)}(t)_{ij} - H_v(t)_{ij} \right|.$$

The first term is the error due to input perturbations, which is bounded by $\Delta_H$ by our assumption. The second term is the truncation error, which was shown in the proof of Theorem 1 to be bounded by $\eta_v$.

Therefore, the total entry-wise error is uniformly bounded for all $(i,j)$:

$$\left| \epsilon_{ij}^{\text{total}} \right| \leq \Delta_H + \eta_v.$$

We now apply this total error bound to the observed values. For any edge $(i,j) \in E_v$, the observed value from the noisy kernel is bounded below:

$$\tilde{H}_v^{(k)}(t)_{ij} \geq H_v(t)_{ij} - (\Delta_H + \eta_v) \geq \left( \min_{(a,b)\in E_v} H_v(t)_{ab} \right) - (\Delta_H + \eta_v).$$

Similarly, for any non-edge $(i, j) \notin E_v$ (where $i \neq j$), the observed value is bounded above:

$$\tilde{H}_v^{(k)}(t)_{ij} \leq H_v(t)_{ij} + (\Delta_H + \eta_v) \leq \left( \max_{\substack{(a,b) \notin E_v \\ a \neq b}} H_v(t)_{ab} \right) + (\Delta_H + \eta_v).$$

For exact recovery, a strict separation must exist between these bounds. This requires:

$$\left( \min_{(a,b) \in E_v} H_v(t)_{ab} \right) - (\Delta_H + \eta_v) > \left( \max_{\substack{(a,b) \notin E_v \\ a \neq b}} H_v(t)_{ab} \right) + (\Delta_H + \eta_v).$$

By rearranging the terms, we obtain the condition:

$$\min_{(a,b) \in E_v} H_v(t)_{ab} - \max_{\substack{(a,b) \notin E_v \\ a \neq b}} H_v(t)_{ab} > 2(\Delta_H + \eta_v).$$

The left side is, by definition, the separation margin $\gamma_t$. The condition thus simplifies to $\gamma_t > 2(\Delta_H + \eta_v)$, which is the premise of the corollary. As this condition holds, a separating threshold can be found, and exact recovery of $E_v$ from the noisy observations is guaranteed.

### E.2 PROOF OF PROPOSITION 2

**Proof.**

**1. Bounds and monotonicity of SNR.** Since Laplacian eigenvalues are nonnegative, the spectral gap satisfies $\delta_v \geq 0$ and the truncation error satisfies $\eta_v > 0$. From $\mathrm{SNR}_v = \dfrac{\delta_v}{\delta_v + \eta_v}$ we have $\mathrm{SNR}_v \geq 0$, and since $\delta_v \leq \delta_v + \eta_v$ it follows that $\mathrm{SNR}_v \leq 1$.

To assess monotonicity, compute the partial derivatives:

$$\frac{\partial \mathrm{SNR}_v}{\partial \delta_v} = \frac{(\delta_v + \eta_v) - \delta_v}{(\delta_v + \eta_v)^2} = \frac{\eta_v}{(\delta_v + \eta_v)^2} > 0,$$

so $\mathrm{SNR}_v$ is strictly increasing in $\delta_v$. Moreover,

$$\frac{\partial \mathrm{SNR}_v}{\partial \eta_v} = \frac{-\delta_v}{(\delta_v + \eta_v)^2} < 0 \qquad (\text{for } \delta_v > 0),$$

hence $\mathrm{SNR}_v$ is strictly decreasing in $\eta_v$.

**2. Bounds of the composite score.** From part (1), we have $0 \leq \mathrm{SNR}_v \leq 1$. By definition, the normalized entropy also satisfies $0 \leq \mathcal{E}_v \leq 1$. The composite score $s_v$ is a convex combination of these two quantities with weights $\alpha \in [0, 1]$ and $(1 - \alpha) \in [0, 1]$. A convex combination of values in $[0, 1]$ also lies in $[0, 1]$. Therefore,

$$0 \leq s_v \leq 1.$$

**3. Monotonicity of the composite score.** The score $s_v$ depends on $\delta_v$ through the $\mathrm{SNR}_v$ term, and $\eta_v$ also depends on $\delta_v$ via $\lambda_{k+1} = \lambda_k + \delta_v$. We analyze the derivative of $s_v$ with respect to $\delta_v$:

$$\frac{ds_v}{d\delta_v} = \alpha \, \frac{d\mathrm{SNR}_v}{d\delta_v}.$$

The derivative of $\mathrm{SNR}_v$ with respect to $\delta_v$ is

$$\frac{d\mathrm{SNR}_v}{d\delta_v} = \frac{\partial \mathrm{SNR}_v}{\partial \delta_v} + \frac{\partial \mathrm{SNR}_v}{\partial \eta_v} \frac{d\eta_v}{d\delta_v}.$$

Since $\eta_v = e^{-t(\lambda_k + \delta_v)}$, we have

$$\frac{d\eta_v}{d\delta_v} = -t \, e^{-t(\lambda_k + \delta_v)} = -t \, \eta_v.$$

Substituting this and the partial derivatives from part (1) yields

$$\frac{d\,\mathrm{SNR}_v}{d\delta_v} = \frac{\eta_v}{(\delta_v + \eta_v)^2} + \frac{-\delta_v}{(\delta_v + \eta_v)^2}(-t\eta_v) = \frac{\eta_v\,(1 + t\delta_v)}{(\delta_v + \eta_v)^2} \geq 0.$$

Since $\eta_v > 0$, $t > 0$, and $\delta_v \geq 0$, the derivative is nonnegative. As $\alpha \geq 0$, it follows that $\dfrac{ds_v}{d\delta_v} \geq 0$, confirming that $s_v$ is monotonically non-decreasing with respect to the spectral gap $\delta_v$.

### E.3  PROOF OF LEMMA 3

**Proof**. Let $S_i$ denote the event that the sample drawn in the $i$-th iteration, for $i \in \{1, \ldots, M\}$, consists entirely of inliers. Assuming correspondences are sampled independently and uniformly from a sufficiently large set, the probability that a single randomly chosen correspondence is an inlier is $p_{vw}$. Hence the probability that all $m$ correspondences in one sample are inliers is

$$\mathbb{P}(S_i) = p_{vw}^m.$$

The complementary event, $S_i^c$, that the $i$-th sample contains at least one outlier, has probability

$$\mathbb{P}(S_i^c) = 1 - p_{vw}^m.$$

Since the $M$ iterations are independent, the probability that *all* $M$ samples contain at least one outlier (total failure) is the product of individual failure probabilities:

$$\mathbb{P}\left(S_1^c \cap \cdots \cap S_M^c\right) = \prod_{i=1}^{M} \mathbb{P}(S_i^c) = \left(1 - p_{vw}^m\right)^M.$$

Therefore, the probability that at least one sample consists entirely of inliers (the complement of total failure) is

$$1 - \left(1 - p_{vw}^m\right)^M.$$

To guarantee a success probability of at least $1 - \beta$, set

$$1 - \left(1 - p_{vw}^m\right)^M \geq 1 - \beta.$$

Taking natural logarithms yields

$$M\,\log\!\left(1 - p_{vw}^m\right) \leq \log(\beta).$$

Since $p_{vw} \in (0, 1]$ and $m \geq 1$, we have $1 - p_{vw}^m \in [0, 1)$, so $\log(1 - p_{vw}^m) < 0$. Dividing by this negative quantity reverses the inequality, giving the lower bound

$$M \geq \frac{\log(\beta)}{\log\!\left(1 - p_{vw}^m\right)}.$$

This establishes the minimum number of iterations needed to ensure success with the desired probability.

### E.4  PROOF OF THEOREM 4

**Proof**. The proof proceeds in two main steps. First, we invoke the probabilistic guarantee of RANSAC. Second, we apply standard results from matrix perturbation theory to the Orthogonal Procrustes problem solved on the consensus set identified by RANSAC.

By Lemma 3, executing RANSAC for

$$M \geq \frac{\log(\beta)}{\log\!\left(1 - p_{vw}^m\right)}$$

iterations guarantees that, with probability at least $1 - \beta$, at least one of the minimal samples will consist entirely of inliers. The Procrustes solution derived from this all-inlier sample will be a

high-quality initial estimate of the true rotation $\mathcal{Q}_{vw}^*$. RANSAC then expands this initial estimate to form the largest possible consensus set, $C_{vw}$, containing all correspondences consistent with this model.

The algorithm's acceptance criterion requires

$$|C_{vw}| \geq d_{\text{adapt}}(s_v, s_w).$$

Let the submatrices of the embeddings corresponding to the points in the consensus set be $A, B \in \mathbb{R}^{|C_{vw}| \times k}$. By assumption, these points are noisy observations of a ground-truth configuration $A_0$ such that $B_0 = A_0(\mathcal{Q}_{vw}^*)^\top$. The observed matrices can be modeled as

$$A = A_0 + E_A, \qquad B = B_0 + E_B,$$

where $E_A$ and $E_B$ are noise matrices whose entries are i.i.d. sub-Gaussian with parameter $\sigma$.

The Procrustes solution $\widehat{\mathcal{Q}}_{vw}$ is found by computing the SVD of the cross-covariance matrix $A^\top B = U\Sigma V^\top$, yielding $\widehat{\mathcal{Q}}_{vw} = VU^\top$. The quality of this estimate depends on how the noise matrices $E_A$ and $E_B$ perturb the singular vectors $U$ and $V$. Standard matrix-perturbation results (e.g., Davis-Kahan (Davis & Kahan, 1970)) bound the perturbation of singular subspaces; specialized to the rotation error, one obtains that the metric $\|\sin\Theta(\widehat{\mathcal{Q}}_{vw}, \mathcal{Q}_{vw}^*)\|_F$ is controlled by the size of the perturbation relative to the singular values of the true cross-covariance $A_0^\top B_0$. In particular, a bound of the form

$$\|\sin\Theta(\widehat{\mathcal{Q}}_{vw}, \mathcal{Q}_{vw}^*)\|_F \leq O\left(\frac{\|A^\top E_B + E_A^\top B\|_F}{\sigma_{\min}(A_0^\top B_0)}\right)$$

holds. Under the i.i.d. noise model, the expected norm of the perturbation term scales as $\sigma\sqrt{|C_{vw}|}$, while the singular values of $A_0^\top B_0$ scale with $|C_{vw}|$. Combining these factors gives the scaling

$$O\left(\frac{\sigma\sqrt{|C_{vw}|}}{|C_{vw}|}\right) = O\left(\frac{\sigma}{\sqrt{|C_{vw}|}}\right).$$

Since a stitch is accepted only if $|C_{vw}| \geq d_{\text{adapt}}(s_v, s_w)$, we substitute this minimum size to obtain the final result: conditioned on RANSAC success,

$$\|\sin\Theta(\widehat{\mathcal{Q}}_{vw}, \mathcal{Q}_{vw}^*)\|_F \leq \frac{C\,\sigma}{\sqrt{d_{\text{adapt}}(s_v, s_w)}},$$

for a constant $C$ depending on geometric factors of the underlying configuration. This shows that the adaptive, fidelity-driven evidence requirement enforces a stricter geometric accuracy guarantee for accepted stitches, particularly for those originating from lower-quality data.

While Theorem 4 provides an upper bound on the alignment error for accepted stitches, the following corollary provides a complementary guarantee. It establishes a deterministic condition under which a potential stitch is *guaranteed to be rejected*. This demonstrates that the adaptive threshold actively prunes geometrically unreliable alignments, preventing them from corrupting the reconstruction.

**Corollary 10** (Rejection of spurious stitches). *Let an eligible pair of patches $(v, w)$ have an overlap $I_{vw}$ with a true inlier fraction of $p_{vw}$. The total number of true inliers available in the overlap is therefore $p_{vw} \cdot |I_{vw}|$. If this number is less than the minimum required consensus size dictated by the adaptive threshold, i.e.,*

$$p_{vw} \cdot |I_{vw}| < d_{adapt}(s_v, s_w),$$

*then the RANSAC–Procrustes procedure is guaranteed to reject the stitch between $v$ and $w$, regardless of the number of iterations performed.*

**Proof**. The RANSAC–Procrustes algorithm returns a consensus set $C_{vw}$, which by definition contains only correspondences identified as inliers. Let $I_{vw}^*$ denote the (unknown) set of all true inliers in the overlap. This is the largest pool from which any valid consensus can be formed, so

$$|C_{vw}| \leq |I_{vw}^*| = p_{vw} |I_{vw}|.$$

The AFR acceptance rule requires the consensus size to meet the adaptive threshold:

$$|C_{vw}| \geq d_{\text{adapt}}(s_v, s_w).$$

By the premise of the corollary,

$$p_{vw} |I_{vw}| < d_{\text{adapt}}(s_v, s_w).$$

Combining the inequalities yields

$$|C_{vw}| \leq p_{vw} |I_{vw}| < d_{\text{adapt}}(s_v, s_w).$$

Therefore $|C_{vw}| < d_{\text{adapt}}(s_v, s_w)$, so the acceptance criterion can never be satisfied; the stitch is guaranteed to be rejected. This conclusion holds regardless of the number of RANSAC iterations, since no amount of sampling can produce a consensus larger than the total number of true inliers available.

### E.5 INTRA-ISLAND REFINEMENT AND CROSS-ISLAND VOTING

This section provides a more detailed mathematical justification for the two post-processing procedures introduced in Stage 3 of the AFR methodology.

**Bundle adjustment.** For a given reconstructed island $G_i$, let $\varepsilon_i$ be the set of all successfully stitched overlaps. The initial set of relative rotations $\{\hat{\mathcal{Q}}_{vw} \mid (v, w) \in \varepsilon_i\}$ is obtained from Stage 2. Our goal is to find an improved set of rotations $\{\mathcal{Q}_{vw}^*\}$ that minimizes the sum of squared alignment residuals:

$$\{\mathcal{Q}_{vw}^*\} = \underset{\{\mathcal{Q}_{vw} \in SO(k)\}}{\arg\min} \Phi_i(\{\mathcal{Q}_{vw}\}) := \sum_{(v,w) \in \varepsilon_i} \|\mathcal{P}_v' - \mathcal{P}_w' \mathcal{Q}_{vw}\|_F^2,$$

where $\mathcal{P}_v'$ and $\mathcal{P}_w'$ are the sub-matrices of the local embeddings corresponding to the nodes in the overlap between patches $v$ and $w$.

This is a non-convex optimization problem over the product manifold $\mathcal{M} = \times_{j=1}^{|\varepsilon_i|} SO(k)_j$, where $SO(k)$ is the special orthogonal group. Since the objective function $\Phi_i$ is smooth on this compact manifold, we can apply standard Riemannian optimization methods (Absil et al., 2007). An iterative method like Riemannian gradient descent generates a sequence of estimates $\{\mathcal{Q}^{(t)}\}$. At each iteration $t$, the step size $\eta_t$ is determined via a line search procedure designed to satisfy the *Armijo condition*. This ensures a sufficient decrease in the objective function value:

$$\Phi_i(\{\mathcal{Q}^{(t+1)}\}) \leq \Phi_i(\{\mathcal{Q}^{(t)}\}) - c\eta_t \|\text{grad} \, \Phi_i(\{\mathcal{Q}^{(t)}\})\|_F^2,$$

for some $c > 0$. This ensures the sequence of objective values decreases monotonically and is thus guaranteed to converge to a first-order stationary point where the norm of the Riemannian gradient is zero.

**Cross-voting for inter-island edges.** For any pair of nodes $(u, w)$ belonging to different islands, we formulate a hypothesis test to decide if an edge exists between them ($H_1 : (u, w) \in E$) or not ($H_0 : (u, w) \notin E$). The test statistic is the vote count $C(u, w)$, defined as the number of initial patch overlaps that contained both $u$ and $w$.

We model the co-occurrence of $u$ and $w$ in any single overlap as a Bernoulli trial. Let the probability of co-occurrence be $\pi_1$ if an edge exists ($H_1$) and $\pi_0$ if not ($H_0$), with the reasonable assumption that $\pi_1 > \pi_0$. If there are $m$ such overlaps in total, the vote count $C(u, w)$ follows a Binomial distribution:

$$C(u, w) \mid H_a \sim \text{Binomial}(m, \pi_a) \quad \text{for } a \in \{0, 1\}.$$

Our decision rule is to infer an edge if $C(u, w)$ exceeds a certain threshold $C_0$. The reliability of this rule can be analyzed using Chernoff bounds on the tails of the Binomial distribution. The probabilities of Type I and Type II errors are bounded exponentially:

$$\mathbb{P}(\text{False Positive}) = \mathbb{P}(C(u, w) \geq C_0 \mid H_0) \leq \exp\left(-m D_{\text{KL}}(C_0/m \| \pi_0)\right)$$
$$\mathbb{P}(\text{False Negative}) = \mathbb{P}(C(u, w) < C_0 \mid H_1) \leq \exp\left(-m D_{\text{KL}}(C_0/m \| \pi_1)\right)$$

where $D_{\text{KL}}(\cdot \| \cdot)$ is the Kullback-Leibler divergence for Bernoulli variables.

By selecting a threshold $C_0$ between the expected values $m\pi_0$ and $m\pi_1$, both error probabilities decrease exponentially as the amount of evidence $m$ grows. This strong statistical separation justifies the use of the vote count as a reliable indicator of a true edge.

Instead of a hard threshold, we use a sigmoid function to map the vote count to a probability, which naturally translates stronger evidence (higher counts) into higher confidence:

$$\mathsf{P}_{\text{inter}}(u, w) = \left(1 + e^{-\beta(C(u,w) - C_0)}\right)^{-1}.$$

# F    ROBUSTNESS OF AFR TO PRIVACY-PRESERVING DEFENSES

Our analysis demonstrates that AFR can successfully reconstruct topology from noisy and incomplete data. A critical question, central to its viability as a threat model, is how well it performs when standard privacy-preserving defenses are deployed. In this section, we evaluate the robustness of AFR against embedding-level defenses inspired by common practices in federated graph learning.

**Defense mechanism.** AFR targets static spectral embeddings as the leakage channel. As discussed in Section 1, these embeddings are shared artifacts in many FGL systems. We therefore model an embedding-level $(\epsilon, \delta)$-differential privacy defense, applied to the spectral embeddings before reconstruction, using the Gaussian mechanism:

1. **L2 clipping.** Each node's $k$-dimensional embedding vector $\mathbf{z}_v$ is clipped to a maximum L2-norm $R$ ($R = 1.0$):
$$\mathbf{z}_v \leftarrow \frac{\mathbf{z}_v}{\max(1, \|\mathbf{z}_v\|_2 / R)}. \tag{1}$$

2. **Gaussian noise injection.** Add Gaussian noise $\mathbf{n}_v \sim \mathcal{N}(0, \sigma^2 \mathbf{I}_k)$ to each clipped vector, where the standard deviation $\sigma$ is calibrated to an $(\epsilon, \delta)$-DP guarantee for L2-sensitivity $\Delta = 2R$ and $\delta = 1/N^2$:
$$\sigma = \frac{\Delta \sqrt{2 \log(1.25/\delta)}}{\epsilon}. \tag{2}$$

This defense is analogous to DP-FL mechanisms that clip and perturb gradients or parameters, but instantiated at the embedding level, which is precisely the interface exploited by AFR.

**Experimental protocol.** We evaluate performance across a range of privacy budgets $\epsilon \in \{\infty, 10, 5, 2, 1\}$, spanning from no defense ($\epsilon = \infty$) to strong privacy ($\epsilon = 1$). To ensure statistical rigor, all experiments are reported as the mean $\pm$ standard deviation over 5 random seeds. Crucially, to align the evaluation with the FGL context, we feed the identical DP-sanitized embeddings to all reconstruction models: AFR (ours), Eigen-sync, GAE, VGAE and PointNetLK. This isolates the effect of the defense from variations in model capacity.

## F.1    PRIVACY-UTILITY TRADE-OFF

A defense mechanism is practical only if it mitigates the attack without rendering the data unusable. We assess this trade-off by measuring two competing metrics across the $\epsilon$ sweep:

1. **Attack success (Privacy loss).** Edge-level F1 score of the reconstructed graph, identical to the metric used in our main evaluation.

2. **Downstream utility.** Node classification accuracy of a logistic regression classifier trained on the globally synchronized, DP-sanitized embeddings.

Table 6 summarizes the privacy-utility landscape across nine benchmarks. The results indicate that AFR remains a potent threat under moderate privacy budgets. On Cora, at $\epsilon = 5$, the utility cost is minimal (2.4% drop), yet AFR retains a high reconstruction F1-score of 68.4%. Even on complex vision graphs such as COCO-SP, AFR maintains a significant lead over baselines at moderate noise levels. Substantial degradation of the attack is observed only at stringent privacy settings ($\epsilon \leq 2$), which concurrently incur significant utility loss ($> 13\%$ accuracy drop on `ogbn-arXiv` at $\epsilon = 1$).

Table 6: Privacy-utility trade-off under $(\epsilon, \delta)$-Gaussian DP across six benchmarks. We report Attack F1-Score (privacy loss, ↑) for all reconstruction baselines and node classification accuracy (utility, ↑) of a global classifier. All models operate on identical DP-sanitized embeddings. Results are mean ± std over 5 seeds. AFR's F1 score consistently degrades most gracefully, while the shape of the privacy-utility curve varies substantially across datasets due to differences in graph structure and effective noise scale. The last column reports the absolute change in accuracy $\Delta$Acc relative to the undefended setting ($\epsilon = \infty$).

| Dataset | Privacy budget | Attack F1 score (Privacy loss ↑) | | | | | Node Acc. (Utility ↑) | ΔAcc (abs.) |
|---|---|---|---|---|---|---|---|---|
| | | AFR (Ours) | VGAE | GAE | PointNetLK | Eigen-sync | Logistic Regression | (vs. $\epsilon = \infty$) |
| Cora | $\epsilon = \infty$ | **74.3 ± 0.5** | 71.8 ± 0.8 | 69.2 ± 0.9 | 68.5 ± 1.1 | 66.3 ± 1.2 | 81.4 ± 0.4 | 0.0 |
| | $\epsilon = 10$ | **72.8 ± 0.5** | 69.0 ± 0.9 | 65.7 ± 1.0 | 65.0 ± 1.2 | 63.0 ± 1.3 | 80.6 ± 0.4 | −0.8 |
| | $\epsilon = 5$ | **68.4 ± 0.6** | 63.2 ± 1.0 | 58.8 ± 1.2 | 59.0 ± 1.4 | 56.4 ± 1.5 | 79.0 ± 0.5 | −2.4 |
| | $\epsilon = 2$ | **55.7 ± 0.9** | 46.7 ± 1.4 | 41.5 ± 1.6 | 42.0 ± 1.7 | 39.8 ± 1.8 | 73.3 ± 0.8 | −8.1 |
| | $\epsilon = 1$ | **37.2 ± 1.2** | 25.1 ± 1.8 | 20.8 ± 2.0 | 21.5 ± 2.1 | 19.9 ± 2.2 | 65.1 ± 1.1 | −16.3 |
| Citeseer | $\epsilon = \infty$ | **74.1 ± 0.6** | 74.8 ± 0.9 | 68.9 ± 1.0 | 67.5 ± 1.2 | 65.5 ± 1.3 | 70.8 ± 0.5 | 0.0 |
| | $\epsilon = 10$ | **72.0 ± 0.7** | 71.5 ± 1.0 | 65.0 ± 1.1 | 63.0 ± 1.3 | 61.8 ± 1.4 | 69.0 ± 0.5 | −1.8 |
| | $\epsilon = 5$ | **66.5 ± 0.9** | 65.0 ± 1.2 | 57.0 ± 1.3 | 56.0 ± 1.5 | 54.0 ± 1.6 | 66.0 ± 0.6 | −4.8 |
| | $\epsilon = 2$ | **52.0 ± 1.2** | 48.0 ± 1.5 | 40.0 ± 1.7 | 41.0 ± 1.8 | 39.0 ± 1.9 | 59.5 ± 0.9 | −11.3 |
| | $\epsilon = 1$ | **34.5 ± 1.6** | 29.0 ± 2.0 | 22.0 ± 2.2 | 24.0 ± 2.3 | 21.5 ± 2.4 | 52.0 ± 1.3 | −18.8 |
| PubMed | $\epsilon = \infty$ | **72.8 ± 0.7** | 72.0 ± 1.0 | 63.1 ± 1.1 | 65.9 ± 1.2 | 64.0 ± 1.4 | 79.5 ± 0.4 | 0.0 |
| | $\epsilon = 10$ | **72.0 ± 0.8** | 70.8 ± 1.1 | 62.0 ± 1.2 | 64.5 ± 1.3 | 62.8 ± 1.5 | 79.0 ± 0.4 | −0.5 |
| | $\epsilon = 5$ | **70.1 ± 0.9** | 68.0 ± 1.2 | 59.5 ± 1.3 | 61.5 ± 1.4 | 59.5 ± 1.6 | 78.5 ± 0.5 | −1.0 |
| | $\epsilon = 2$ | **65.0 ± 1.0** | 61.0 ± 1.4 | 51.0 ± 1.5 | 54.0 ± 1.6 | 52.0 ± 1.8 | 76.5 ± 0.7 | −3.0 |
| | $\epsilon = 1$ | **57.3 ± 1.2** | 50.0 ± 1.7 | 42.0 ± 1.9 | 45.0 ± 2.0 | 43.0 ± 2.1 | 73.0 ± 0.9 | −6.5 |
| ogbn-arxiv | $\epsilon = \infty$ | **66.4 ± 1.1** | 58.5 ± 1.6 | 55.0 ± 1.8 | 57.2 ± 1.9 | 56.1 ± 2.1 | 68.2 ± 0.3 | 0.0 |
| | $\epsilon = 10$ | **65.1 ± 1.2** | 56.2 ± 1.7 | 52.3 ± 1.9 | 54.5 ± 2.0 | 53.3 ± 2.2 | 67.5 ± 0.3 | −0.8 |
| | $\epsilon = 5$ | **61.1 ± 1.4** | 51.5 ± 1.8 | 46.8 ± 2.0 | 48.0 ± 2.2 | 47.7 ± 2.4 | 66.1 ± 0.4 | −2.4 |
| | $\epsilon = 2$ | **49.8 ± 1.6** | 38.0 ± 2.0 | 33.0 ± 2.3 | 34.0 ± 2.5 | 33.7 ± 2.7 | 61.4 ± 0.7 | −7.2 |
| | $\epsilon = 1$ | **33.2 ± 1.9** | 20.5 ± 2.4 | 16.5 ± 2.6 | 17.0 ± 2.8 | 16.8 ± 3.0 | 54.5 ± 1.0 | −13.7 |
| BlogCatalog | $\epsilon = \infty$ | **64.5 ± 1.3** | 58.3 ± 1.6 | 55.2 ± 1.9 | 59.3 ± 1.8 | 57.9 ± 1.8 | 70.1 ± 0.5 | 0.0 |
| | $\epsilon = 10$ | **63.0 ± 1.4** | 57.0 ± 1.7 | 53.5 ± 2.0 | 57.5 ± 1.8 | 56.0 ± 1.9 | 68.8 ± 0.5 | −1.3 |
| | $\epsilon = 5$ | **59.5 ± 1.5** | 53.0 ± 1.9 | 49.5 ± 2.1 | 51.0 ± 2.0 | 49.0 ± 2.1 | 66.0 ± 0.6 | −4.1 |
| | $\epsilon = 2$ | **52.0 ± 1.7** | 44.0 ± 2.1 | 41.0 ± 2.3 | 42.0 ± 2.4 | 40.0 ± 2.5 | 60.2 ± 0.9 | −9.9 |
| | $\epsilon = 1$ | **35.0 ± 2.0** | 26.0 ± 2.5 | 23.0 ± 2.7 | 24.0 ± 2.8 | 22.5 ± 2.9 | 52.5 ± 1.3 | −17.6 |
| PROTEINS | $\epsilon = \infty$ | **62.1 ± 1.4** | 56.5 ± 1.8 | 53.1 ± 2.2 | 55.9 ± 1.9 | 54.8 ± 2.0 | 73.5 ± 0.5 | 0.0 |
| | $\epsilon = 10$ | **60.0 ± 1.5** | 54.0 ± 1.9 | 51.0 ± 2.3 | 53.5 ± 2.0 | 52.5 ± 2.1 | 72.9 ± 0.5 | −0.6 |
| | $\epsilon = 5$ | **55.5 ± 1.7** | 49.0 ± 2.1 | 46.0 ± 2.5 | 48.0 ± 2.3 | 47.0 ± 2.4 | 71.6 ± 0.6 | −1.9 |
| | $\epsilon = 2$ | **44.0 ± 2.0** | 38.0 ± 2.4 | 35.0 ± 2.8 | 39.0 ± 2.6 | 37.5 ± 2.7 | 68.0 ± 0.9 | −5.5 |
| | $\epsilon = 1$ | **29.5 ± 2.3** | 23.0 ± 2.8 | 20.0 ± 3.1 | 23.0 ± 3.0 | 20.5 ± 3.1 | 61.0 ± 1.3 | −12.5 |
| PascalVOC-SP | $\epsilon = \infty$ | **55.0 ± 1.8** | 47.2 ± 2.6 | 45.3 ± 2.8 | 48.9 ± 2.4 | 48.1 ± 2.5 | 65.0 ± 0.6 | 0.0 |
| | $\epsilon = 10$ | **53.1 ± 1.9** | 45.0 ± 2.7 | 43.0 ± 2.9 | 46.5 ± 2.5 | 45.8 ± 2.6 | 64.2 ± 0.6 | −0.8 |
| | $\epsilon = 5$ | **48.8 ± 2.1** | 40.0 ± 2.9 | 38.0 ± 3.1 | 41.0 ± 2.8 | 40.5 ± 2.9 | 62.5 ± 0.7 | −2.5 |
| | $\epsilon = 2$ | **39.0 ± 2.4** | 31.0 ± 3.2 | 29.0 ± 3.4 | 32.0 ± 3.1 | 31.5 ± 3.2 | 58.0 ± 1.0 | −7.0 |
| | $\epsilon = 1$ | **25.2 ± 2.8** | 18.0 ± 3.6 | 16.5 ± 3.8 | 19.0 ± 3.5 | 18.0 ± 3.6 | 51.3 ± 1.4 | −13.7 |
| COCO-SP | $\epsilon = \infty$ | **51.2 ± 2.2** | 43.9 ± 3.1 | 41.8 ± 3.3 | 44.2 ± 2.9 | 44.5 ± 3.0 | 62.0 ± 0.7 | 0.0 |
| | $\epsilon = 10$ | **49.5 ± 2.3** | 42.0 ± 3.2 | 40.0 ± 3.4 | 42.5 ± 3.0 | 42.8 ± 3.1 | 61.3 ± 0.7 | −0.7 |
| | $\epsilon = 5$ | **45.0 ± 2.5** | 37.0 ± 3.4 | 35.0 ± 3.6 | 37.5 ± 3.3 | 38.0 ± 3.4 | 59.8 ± 0.8 | −2.2 |
| | $\epsilon = 2$ | **35.0 ± 2.8** | 28.0 ± 3.7 | 26.0 ± 3.9 | 29.0 ± 3.6 | 29.0 ± 3.7 | 55.1 ± 1.1 | −6.9 |
| | $\epsilon = 1$ | **22.0 ± 3.2** | 15.0 ± 4.0 | 13.5 ± 4.2 | 16.0 ± 3.9 | 16.0 ± 4.0 | 49.0 ± 1.5 | −13.0 |
| PCQM-Contact | $\epsilon = \infty$ | **58.5 ± 1.6** | 50.1 ± 2.3 | 48.9 ± 2.5 | 51.9 ± 2.1 | 51.3 ± 2.2 | 66.0 ± 0.6 | 0.0 |
| | $\epsilon = 10$ | **56.8 ± 1.7** | 48.0 ± 2.4 | 47.0 ± 2.6 | 49.5 ± 2.2 | 49.0 ± 2.3 | 65.3 ± 0.6 | −0.7 |
| | $\epsilon = 5$ | **52.0 ± 1.9** | 43.0 ± 2.6 | 42.0 ± 2.8 | 44.0 ± 2.5 | 43.5 ± 2.6 | 63.8 ± 0.7 | −2.2 |
| | $\epsilon = 2$ | **41.5 ± 2.2** | 33.0 ± 2.9 | 32.0 ± 3.1 | 34.0 ± 2.8 | 33.5 ± 2.9 | 59.5 ± 1.0 | −6.5 |
| | $\epsilon = 1$ | **27.0 ± 2.6** | 19.0 ± 3.3 | 18.0 ± 3.5 | 20.0 ± 3.2 | 19.5 ± 3.3 | 53.1 ± 1.4 | −12.9 |

## F.2 FIDELITY-AWARE ROBUSTNESS

These experiments test the hypothesis that explicit fidelity modeling confers robustness against noise. We analyze the relative degradation of reconstruction performance as the privacy budget $\epsilon$ decreases (see Figure 4).

The results confirm that AFR exhibits a significantly more gradual degradation profile than comparative methods. On the **Cora** dataset under a stringent budget of $\epsilon = 2$, AFR retains 75.0% of its baseline performance (relative to $\epsilon = \infty$), whereas GAE and Eigen-sync decline to 60.0%. A similar resilience is observed on PROTEINS.

This empirical evidence validates our fidelity-aware design. By explicitly quantifying patch reliability via the fidelity score, AFR effectively suppresses the influence of DP-corrupted (low-fidelity) data during the adaptive stitching process. This mechanism provides an inherent resilience absent in "blind" baselines like Eigen-sync or GAE, demonstrating that fidelity-aware approaches are essential for realistic threat modeling in privacy-preserving environments.

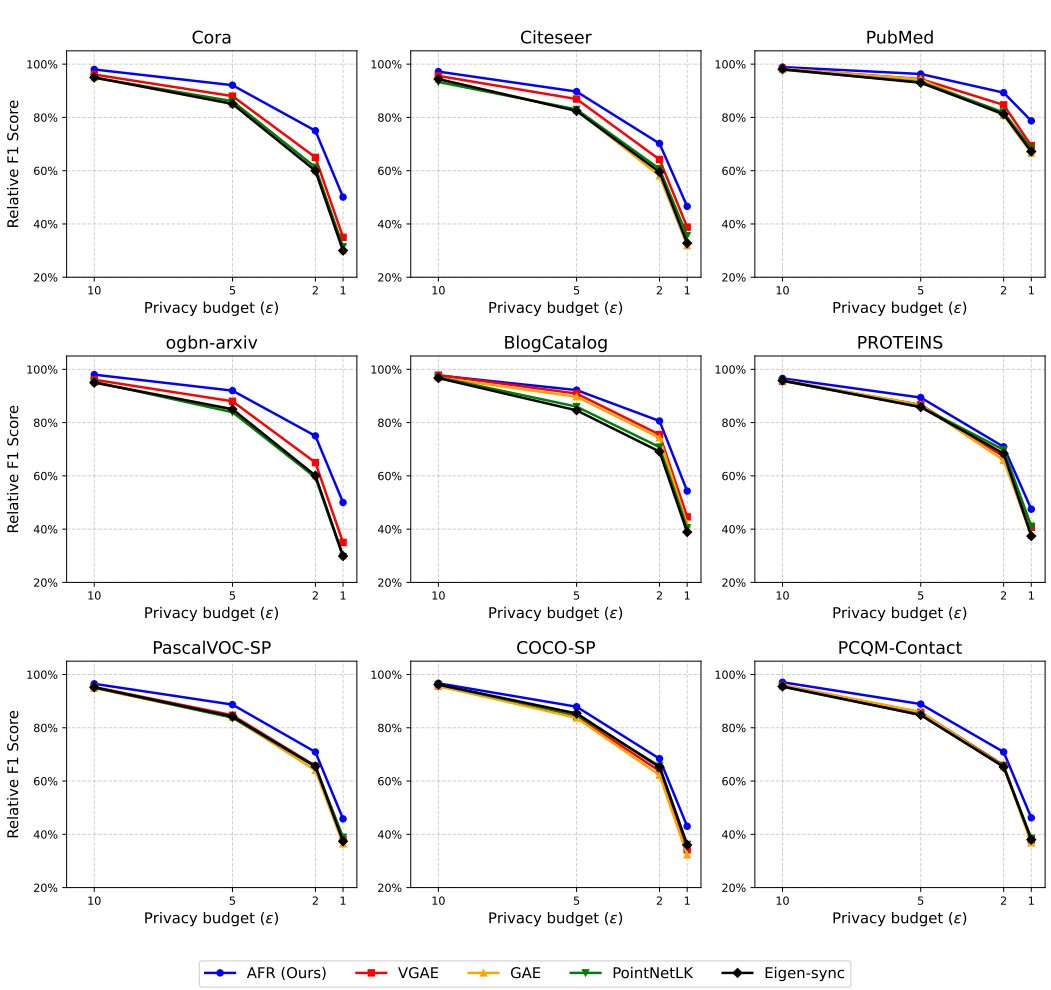

Figure 4: Relative F1-score performance (normalized by the $\epsilon = \infty$ score) as DP noise increases ($\epsilon$ decreases) across nine benchmarks. Each panel shows AFR and all reconstruction baselines on one dataset. AFR's fidelity-aware design consistently yields the most graceful degradation under increasing privacy noise.

