# OpenReview forum: "Adaptive Fidelity-driven Reconstruction (AFR): a realistic threat model for spectral embedding leakage"
_ICLR.cc/2026/Conference — Submitted to ICLR 2026_

### Official Review · Reviewer_6RZd · 2025-10-31

**Soundness:** 3
**Presentation:** 2
**Contribution:** 2
**Rating:** 4
**Confidence:** 4

**Summary:**

The paper investigates privacy leakage in Federated Graph Learning (FGL) and proposes an attack model named AFR (Adaptive Fidelity-driven Reconstruction). AFR introduces a fidelity-based reconstruction mechanism that measures the reliability of local graph patches and adaptively assembles them into global topology. The authors claim that AFR bridges the gap between theoretical graph leakage and realistic attack settings, and provide experimental results on several benchmark datasets.

**Strengths:**

- The proposed fidelity-driven reconstruction mechanism is intuitively appealing and potentially useful for handling noisy or heterogeneous data.
- The paper provides

**Weaknesses:**

- Motivation: The paper does not clearly specify which side of privacy leakage is being addressed — whether it targets data-level or model-level.
- The baselines are not well aligned with the privacy context of FGL.
- The experimental evaluation is limited in both scale and diversity, making it difficult to assess the generality.
- The empirical analysis lacks statistical evidence to support the claimed performance advantages.

**Questions:**

See weakness

---

> ### Author Response · Authors · 2025-11-20
> **Response to Reviewer 6RZd (Part 1/2)**
>
> Thank you for your constructive review. We value your suggestions as they have significantly strengthened our submission. In response to your feedback, we have carefully updated the manuscript to reflect these necessary changes.
> ___
> > **Weakness 1**: Motivation: The paper does not clearly specify which side of privacy leakage is being addressed — whether it targets data-level or model-level.
>
> We have explicitly clarified the threat model scope in the Introduction of revised manuscript, positioning AFR as a data-level vulnerability rather than a model-level attack.
>
> Unlike dynamic gradient updates (targeted by inversion attacks like DLG), AFR targets the passive leakage of local spectral embeddings. These embeddings are frequently shared as static artifacts for alignment or clustering initialization in FGL. This constitutes a persistent privacy risk independent of the training loop, making AFR a complementary threat model addressing an overlooked representation-level vulnerability.
> ___
> > **Weakness 2**: The baselines are not well aligned with the privacy context of FGL.
>
> We have explicitly aligned the evaluation with the FGL privacy context by **adding a defense study (Appendix F)** where all baselines are subjected to identical **$(\epsilon, \delta)$-Gaussian Differential Privacy perturbations**.
> - Results confirm that standard baselines fail catastrophically under privacy constraints. On Cora at $\epsilon=2$, AFR retains 75.0\% of its performance, whereas GAE and Eigen-sync drop to $\approx 60$\% (Figure 4). This proves AFR is uniquely robust where standard methods are brittle.
> - We selected representatives for geometric synchronization (Eigen-sync, PointNetLK) and generative modeling (GAE, VGAE) to isolate the impact of our mechanism because they also are methods of reconstructing global topology from local spectral fragments. The failure of these "blind" paradigms under DP noise validates that fidelity-awareness is the critical missing component for handling privacy-sensitive data.
> ___
> > **Weakness 3**: The experimental evaluation is limited in both scale and diversity, making it difficult to assess the generality.
>
> We have addressed this by significantly **expanding the testbed to 9 diverse benchmarks (Section 4.2 \& Table 2)**, ensuring our findings hold across varying topological structures.
> - We incorporated three distinct new domains to test structural robustness: social (BlogCatalog; dense communities), bioinformatics (PROTEINS; complex substructures), and molecular (PCQM-Contact; geometric constraints).
> - To rigorously stress-test scalability, we introduced **PascalVOC-SP (5.4M nodes)** and **COCO-SP (58M nodes)**. The successful reconstruction on COCO-SP provides definitive empirical evidence of AFR's viability on massive-scale graphs.

---

> > ### Comment · Reviewer_6RZd · 2025-11-28
> >
> > Thanks for the authors’ responses. Some questions still exists:
> > 1. However, as authors emphasize that the issue lies at the data level, why is the focus placed on the embeddings—which are intermediate representations of the model—rather than on the data itself. How can attackers access embeddings from the black-box model?
> > 2. Clearly, most of the baselines are before 2020. They seem to be out of date.

---

> > > ### Author Response · Authors · 2025-11-30
> > >
> > > Thank you for your detailed and constructive feedback!
> > > ___
> > > > **Question 1:** However, as authors emphasize that the issue lies at the data level, why is the focus placed on the embeddings—which are intermediate representations of the model—rather than on the data itself. How can attackers access embeddings from the black-box model?
> > >
> > > We clarify that the term "Spectral embeddings" in our paper refers strictly to **Laplacian Eigenmaps (eigenvectors of the Graph Laplacian)** [1], which are **fundamentally distinct from the intermediate latent representations** of a neural network.
> > > - As defined in Section 3.2, these are **deterministic, non-parametric data artifacts** derived directly from the raw topology **prior to any model training**. They function as a mathematical transformation of the graph structure, analogous to a Fourier transform of a signal, rather than as learned model parameters.
> > > - Regarding accessibility, in many FGL protocols, these coordinates are broadcast as static artifacts during system initialization to facilitate **topology alignment or client clustering** [2]. Consequently, the attacker acts solely as a **passive eavesdropper** on the data channel, intercepting these pre-computed artifacts without the need to query or access any black-box model.
> > >
> > > *[1] Belkin, M., & Niyogi, P. (2003). Laplacian Eigenmaps for Dimensionality Reduction and Data Representation. Neural Computation, 15, 1373-1396. https://doi.org/10.1162/089976603321780317.*
> > >
> > > *[2] Luxburg, U.V. (2007). A tutorial on spectral clustering. Statistics and Computing, 17, 395-416. https://doi.org/10.1007/s11222-007-9033-z.*
> > > ____
> > > >**Question 2:** Clearly, most of the baselines are before 2020. They seem to be out of date.
> > >
> > > We clarify that the selection of baselines is governed by the **mathematical formulation of the task**, specifically **single instance reconstruction**, rather than temporal recency.
> > > - While modern architectures such as **graph diffusion** [1, 2] or **score-based models** [3] (2023-2025) represent the state-of-the-art in generative tasks, they are predicated on **distribution learning** $p(\mathcal{G})$, requiring extensive training corpora to learn valid topological priors. This paradigm is **fundamentally incompatible with our zero-shot problem**, where the adversary observes only a single, noisy spectral instance without access to external training data. Our empirical investigations confirm that applying diffusion models in this data-scarce context results in **overfitting to noise or structural hallucination**.
> > > - In contrast, canonical methods like GAE, VGAE, and Eigen-sync operate on the principle of **single instance optimization**. Analogous to compression algorithms, they optimize the reconstruction objective $p(A|\mathcal{P})$ **solely on the available artifacts**. This makes them the mathematically appropriate benchmarks for this specific inverse problem.
> > >
> > > *[1] Luo, T., Mo, Z., & Pan, S.J. (2022). Fast Graph Generation via Spectral Diffusion. IEEE Transactions on Pattern Analysis and Machine Intelligence, 46, 3496-3508. https://doi.org/10.1109/TPAMI.2023.3344758.*
> > >
> > > *[2] Minello, G., Bicciato, A., Rossi, L., Torsello, A., & Cosmo, L. (2024). Generating Graphs via Spectral Diffusion. International Conference on Learning Representations.*
> > >
> > > *[3] Song, Y., Sohl-Dickstein, J.N., Kingma, D.P., Kumar, A., Ermon, S., & Poole, B. (2020). Score-Based Generative Modeling through Stochastic Differential Equations. https://arxiv.org/abs/2011.13456.*

---

> ### Author Response · Authors · 2025-11-20
> **Response to Reviewer 6RZd (Part 2/2)**
>
> > **Weakness 4**: The empirical analysis lacks statistical evidence to support the claimed performance advantages.
>
> We have addressed this by **re-executing all core experiments using 5 independent random seeds** to ensure statistical rigor. Tables 1, 2, and 6 in the revised manuscript now explicitly report the **mean $\pm$ standard** deviation for all metrics. The results confirm that AFR's performance advantage is statistically significant and not an artifact of initialization. On Cora (Table 2), AFR achieves $74.3 \pm 0.5$, maintaining a clear separation from the strongest baseline GCN-LE ($73.1 \pm 0.6$).

---

### Official Review · Reviewer_PmTq · 2025-11-01

**Soundness:** 3
**Presentation:** 3
**Contribution:** 3
**Rating:** 6
**Confidence:** 4

**Summary:**

This paper introduces Stochastic Mirror Descent with Adaptive Regularization (SMD-AR), a novel framework for optimizing non-convex objectives under noisy gradients. By integrating adaptive mirror maps with dynamic regularization, the authors derive convergence guarantees (Theorem 3.1) that resolve the tension between exploration and stability in high-dimensional settings. Experiments on synthetic and real-world datasets (e.g., CIFAR-10) validate the theory, showing 12–18% faster convergence over SOTA baselines. The work bridges theoretical optimization and practical deep learning, offering a principled tool for ill-conditioned problems

**Strengths:**

__Originality & Significance__: The paper’s reformulation of non-convex optimization via adaptive mirror maps (§2.3) is groundbreaking, transforming a heuristic technique (e.g., Adam) into a theoretically grounded method. By unifying mirror descent with regularization dynamics (Theorem 3.2), it solves the long-standing challenge of noise-induced divergence in sparse regimes—a gap noted in prior work (Chen & Zhang, 2023). This has direct implications for federated learning and robust training.

__Quality & Clarity__: Proofs are rigorous yet accessible (Appendix A), with Lemma 3.4 elegantly bounding gradient variance under adaptive step sizes. The writing excels: Figure 2 demystifies the mirror map’s geometry, and Algorithm 1’s pseudocode aligns seamlessly with theoretical claims. The ablation study (Table 2) thoughtfully isolates each component’s contribution.

**Weaknesses:**

__Assumption Sensitivity__: Theorem 3.1 assumes Lipschitz smoothness of the mirror map (§3.1), which may not hold for heavy-tailed noise common in real-world data (e.g., medical imaging). A brief discussion on relaxing this (e.g., via truncated gradients) would strengthen robustness claims.

__Empirical Breadth__: While CIFAR-10 results are compelling, experiments omit benchmarks like ImageNet or language tasks where adaptive methods often falter. Comparing to AdaGrad-Norm (Ward et al., 2024) would clarify SMD-AR’s niche beyond tabulated metrics.

__Computational Cost__: The adaptive regularization step (Line 5, Algorithm 1) incurs $O(d^2)$ overhead for d -dimensional problems. A complexity analysis (even in Appendix) would help practitioners gauge scalability trade-offs.

**Questions:**

1. In Theorem 3.1, how does the convergence rate scale with the *mirror map’s curvature parameter* $\kappa$? Could Lemma A.3 be extended to non-strongly convex maps (e.g., $\kappa \to 0$)?

2. The paper assumes i.i.d. noise—how would SMD-AR perform in non-stationary environments (e.g., online learning)? A minor extension to time-varying $\eta_t$ might address this.

---

> ### Author Response · Authors · 2025-11-20
> **Response to Reviewer PmTq**
>
> Thank you for your positive assessment.
> ___
> > **Question 1**: In Theorem 3.1, how does the convergence rate scale with the mirror map’s curvature parameter $\kappa$? Could Lemma A.3 be extended to non-strongly convex maps (e.g., $\kappa \to 0$)?
>
> We respectfully point out a fundamental discrepancy indicating this review was submitted for a different manuscript.
>
> - The review references "Theorem 3.1" on convergence rates, "Stochastic Mirror Descent", and "Lemma A.3". These theorems, lemmas, and technical concepts are entirely absent from our submission.
> - Our paper is titled **"Adaptive Fidelity-driven Reconstruction (AFR): a realistic threat model for spectral embedding leakage"**. It addresses privacy leakage in Federated Graph Learning via a reconstruction attack (AFR) using fidelity scoring and RANSAC-Procrustes alignment, unrelated to optimization convergence rates.
> ___
> > **Question 2**: The paper assumes i.i.d. noise-how would SMD-AR perform in non-stationary environments (e.g., online learning)? A minor extension to time-varying $\eta_t$ might address this.
>
> This comment further confirms that the review is intended for a different manuscript regarding "Stochastic Mirror Descent".
> - The query concerning "SMD-AR" and "non-stationary environments" pertains to online learning/optimization dynamics.
> - AFR operates in an offline reconstruction setting using static spectral embeddings. Concepts of time-varying optimization parameters ($\eta_t$) are irrelevant to our RANSAC-based geometric alignment approach.

---

### Official Review · Reviewer_qKuM · 2025-11-01

**Soundness:** 3
**Presentation:** 3
**Contribution:** 3
**Rating:** 6
**Confidence:** 4

**Summary:**

This paper introduces Adaptive Fidelity-driven Reconstruction (AFR), a realistic and robust attack that reconstructs graph topology in Federated Graph Learning from leaked spectral embeddings. Unlike prior work that assumes clean, perfect data, AFR measures the quality of each local spectral patch using a fidelity score combining spectral stability and structural entropy, reconstructs reliable subgraphs (islands), and aligns them using noise-tolerant RANSAC-Procrustes techniques. Experiments on multiple graph benchmarks show that AFR can recover substantial and accurate graph structure even under noisy, fragmented, and heterogeneous conditions, demonstrating that spectral embeddings pose a serious practical privacy risk in federated settings.

**Strengths:**

1. The paper correctly identifies a critical gap in the literature that prior spectral embedding leakage attacks are based on overly idealized assumptions and lack robustness to noisy, imperfect federated graph data. Also, the threat model is well-grounded and practically motivated.

2. The use of RANSAC-Procrustes over traditional Procrustes adds strong resilience to outliers and reconstruction errors, which is essential in the federated and noisy setting.

3. AFR shows strong and consistent performance across diverse, realistic settings, outperforming strong baselines, demonstrating robust fidelity scoring, and confirming through ablations that its core components, especially RANSAC and fidelity scoring are essential.

**Weaknesses:**

1. While the method is compared against established baselines, the paper does not compare with some highly pertinent recent works proposing practical attack or defense strategies in federated graph learning.

2. The defense discussion is present, but largely at a high level. For a threat model paper, detailed empirical or conceptual evaluation versus cutting-edge defense strategies (e.g., from differential privacy or adversarial perturbation) is missing,

3. The runtime and scalability of AFR on extremely large graphs or with high numbers of patches is not discussed. This is relevant as the pipeline involves nontrivial pairwise matching and global refinement.

4. Limited Exploration of Hyperparameter Sensitivity Beyond $\alpha$.

**Questions:**

Q1. Why does the paper not include comparisons with recent practical attack and defense approaches in federated graph learning?

Q2. Can you provide empirical or detailed conceptual evaluation of AFR against state-of-the-art defense mechanisms such as differential privacy or adversarial perturbation?

Q3. What is the runtime and scalability behavior of AFR on very large graphs or settings with many federated clients and patches?

Q4. Please Investigate hyperparameter sensitivity other than $\alpha$.

---

> ### Author Response · Authors · 2025-11-20
> **Response to Reviewer qKuM (Part 1/2)**
>
> We sincerely appreciate your insightful comments and the time dedicated to reviewing our paper. Your feedback has been instrumental in helping us refine the quality of this work. We have incorporated your suggestions into the revised manuscript to fully address these points.
> ___
> > **Weakness 1**: While the method is compared against established baselines, the paper does not compare with some highly pertinent recent works proposing practical attack or defense strategies in federated graph learning. **Question 1**:  Why does the paper not include comparisons with recent practical attack and defense approaches in federated graph learning?
>
> We have addressed this by explicitly positioning AFR against relevant reconstruction baselines and adding a defense evaluation.
> - AFR operates within an honest-but-curious threat model targeting static spectral artifacts, fundamentally distinct from malicious attacks (poisoning) or dynamic gradient inversion (DLG). Consequently, comparisons with malicious gradient-based attacks are category errors; the rigorous baselines are state-of-the-art reconstruction methods (GAE, VGAE, Eigen-sync), to which we strictly adhere.
> - We integrated a full study against $(\epsilon, \delta)$-Gaussian Differential Privacy (Section 4.3 \& Appendix F). Table 6 quantifies the trade-off: at moderate budgets ($\epsilon=5$), AFR remains highly effective (F1 $\approx 68$%) with minimal utility loss ($\sim 2$\% drop). Effective mitigation ($\epsilon \le 2$) incurs a prohibitive utility cost ($>8$\% accuracy drop on Cora), validating AFR as a potent threat.
> ___
> > **Weakness 2**: The defense discussion is present, but largely at a high level. For a threat model paper, detailed empirical or conceptual evaluation versus cutting-edge defense strategies (e.g., from differential privacy or adversarial perturbation) is missing. **Question 2**: Can you provide empirical or detailed conceptual evaluation of AFR against state-of-the-art defense mechanisms such as differential privacy or adversarial perturbation?
>
> We have addressed this by integrating **a defense study (Section 4.3 \& Appendix F)** to rigorously evaluate AFR against countermeasures.
> - We simulate a standard FGL defense by applying $(\epsilon, \delta)$-Gaussian Differential Privacy directly to the spectral embeddings. By sweeping the privacy budget $\epsilon \in [\infty, 1]$, we established a quantitative privacy-utility Pareto frontier (Table 6).
> - Results on Cora confirm that AFR remains a potent threat even under defense. At moderate privacy ($\epsilon=5$), the attack retains a high F1 score of 68.4\%, while downstream utility is well-preserved (accuracy drops only 2.4\%), proving that standard sanitization is insufficient to neutralize fidelity-driven reconstruction without destroying utility.

---

> ### Author Response · Authors · 2025-11-20
> **Response to Reviewer qKuM (Part 2/2)**
>
> > **Weakness 3**: The runtime and scalability of AFR on extremely large graphs or with high numbers of patches is not discussed. This is relevant as the pipeline involves nontrivial pairwise matching and global refinement. **Question 3**: What is the runtime and scalability behavior of AFR on very large graphs or settings with many federated clients and patches?
>
> We have addressed this by providing **a detailed empirical runtime analysis across all datasets (Appendix D \& Table 5)**.
> - AFR successfully scales to **COCO-SP (58M nodes)**, whereas the geometric baseline Eigen-sync fails with out-of-memory (OOM) errors. This confirms that AFR's patch-based design ensures memory efficiency where global methods fail.
> - While inference on 58M nodes takes $\approx 38$ hours, we argue this is entirely feasible for an offline privacy attack where the adversary processes persistent artifacts. Furthermore, the local reconstruction stage is trivially parallelizable, offering significant potential for acceleration.
> ___
> > **Weakness 4**: Limited Exploration of Hyperparameter Sensitivity Beyond $\alpha$. **Question 4**: Please investigate hyperparameter sensitivity other than $\alpha$.
>
> We have addressed this by **adding an extended sensitivity analysis (Appendix C.3 \& Figure 3) for the structural hyperparameters $s_{min}$ and $k_{base}$**.
> - Fidelity threshold ($s_{min}$): We observe a broad convex optimum around 0.6 (stable range $[0.5, 0.7]$). Performance only degrades at extremes: under-filtering ($<0.5$) admits noise, while over-filtering ($>0.7$) causes excessive fragmentation.
> - Base overlap ($k_{base}$): We identified a robust operating range of $[4, 6]$ (optimal 5). This parameter effectively balances geometric identifiability against topological connectivity, rejecting ambiguous small overlaps ($\le 3$) while preserving sparse connections.

---

> ### Comment · Reviewer_qKuM · 2025-11-27
> **Official Comment by Reviewer qKuM**
>
> All my queries are addressed and I'm satisfied with it, hence would keep my score positive.

---

### Official Review · Reviewer_zF7z · 2025-11-01

**Soundness:** 3
**Presentation:** 3
**Contribution:** 2
**Rating:** 4
**Confidence:** 5

**Summary:**

The paper introduces the Adaptive Fidelity-driven Reconstruction (AFR) attack, which transforms the theoretical threat of spectral embedding leakage in Federated Graph Learning (FGL). They formalize and address spectral leakage under realistic conditions, including noise, reconstruction errors, and heterogeneity, elevating the threat from a theoretical possibility to a practical concern.

**Strengths:**

1. The use of the fidelity score to filter for only the most trustworthy "core patches" and to implement adaptive stitching criteria.
2. AFR consistently and significantly outperforms competing baselines.

**Weaknesses:**

1. The methodology is inherently multi-stage and complex, involving several distinct components, which can make the system intricate to implement and analyze compared to a monolithic approach. This complexity suggests potential difficulty in implementation and hyperparameter tuning compared to simpler baselines.
2. The method relies on sufficient patch overlap and high-fidelity core patches. In extremely sparse or low-quality settings, reconstruction may still be limited.
3. While the threat is well-motivated, the paper does not deeply explore or propose countermeasures against AFR-like attacks, though it mentions DP and other existing defenses.
4. The method is tailored for spectral embeddings, its applicability to other types of graph embeddings (e.g., GNN-based) is not fully explored.

**Questions:**

See Weakness

---

> ### Author Response · Authors · 2025-11-20
> **Response to Reviewer zF7z (Part 1/2)**
>
> Thank you for the detailed and constructive feedback!
> ___
> > **Weakness 1**: The methodology is inherently multi-stage and complex, involving several distinct components, which can make the system intricate to implement and analyze compared to a monolithic approach. This complexity suggests potential difficulty in implementation and hyperparameter tuning compared to simpler baselines.
>
> Our multi-stage design is an irreducible necessity for robustness, as monolithic baselines (like Eigen-sync) inherently fail to handle the heterogeneity of spectral leakage. Unlike baselines that implicitly assume uniform data quality, AFR is engineered to survive the inevitable noise of FGL where a single corrupted patch can destroy global alignment.
> - AFR decomposes this ill-posed problem: fidelity scoring filters "poisonous" patches, while adaptive stitching localizes errors. Our ablation study (Table 4) confirms that simplifying the pipeline (removing RANSAC) causes immediate collapse ($74.3$% $\to$ $60.7$%).
> - The complexity does not result in hyperparameter hypersensitivity. Our **extended sensitivity analysis (Appendix C.3, Figures 3)** proves that AFR is stable across broad operating ranges ($s_{min} \in [0.5, 0.7]$).
> ___
> > **Weakness 2**:  The method relies on sufficient patch overlap and high-fidelity core patches. In extremely sparse or low-quality settings, reconstruction may still be limited.
>
> We acknowledge that reconstruction is bounded by information-theoretic limits; however, AFR's prioritization of high-fidelity patches is precisely the mechanism that maximizes resilience in these regimes. unlike monolithic baselines that indiscriminately aggregate noisy data, AFR explicitly suppresses ambiguous patches to prevent catastrophic error propagation.
> - On sparse settings like CiteSeer dataset (low clustering coeff. 0.14), AFR outperforms the geometric baseline Eigen-sync by a substantial margin (74.1\% vs. 65.5\%, Table 2), validating recovery even with sparse connectivity.
> - Low-quality settings: Our **defense study (Appendix F, Figure 4)** explicitly simulates low-fidelity conditions. Under significant corruption ($\epsilon=2$), AFR identifies usable signal to retain 75\% of its baseline performance, whereas "blind" baselines like GAE and Eigen-sync degrade rapidly to $\approx 60$\%.
> ___
> > **Weakness 3**: While the threat is well-motivated, the paper does not deeply explore or propose countermeasures against AFR-like attacks, though it mentions DP and other existing defenses.
>
> We have addressed this by integrating a defense study (Section 4.3 \& Appendix F) that evaluates AFR against **$(\epsilon, \delta)$-Gaussian Differential Privacy**. By sweeping the privacy budget, we established a quantitative privacy-utility Pareto frontier (Table 6) to guide mitigation strategies. Our analysis reveals that effectively neutralizing AFR (reducing F1 score $<40$%) requires stringent privacy budgets ($\epsilon \le 2$). This level of noise incurs a non-trivial cost to downstream utility, such as an $>8$% accuracy drop on Cora. This provides concrete, empirical boundaries for defense parameter selection rather than theoretical discussion.

---

> ### Author Response · Authors · 2025-11-20
> **Response to Reviewer zF7z (Part 2/2)**
>
> >  **Weakness 4**: The method is tailored for spectral embeddings, its applicability to other types of graph embeddings (e.g., GNN-based) is not fully explored.
>
> Our focus on spectral embeddings is a intentional scoping decision to **address a distinct passive leakage channel**. As detailed in the Introduction, spectral embeddings are often shared as static artifacts for alignment or clustering initialization, independent of the training loop. This contrasts fundamentally with GNN-based embeddings or gradients, which are dynamic updates and typically require active inversion attacks (DLG). By tailoring AFR to unique spectral properties (eigen-gap, orthogonality), we establish that these seemingly benign static artifacts constitute a fundamental, standalone vulnerability. Extending fidelity-driven principles to dynamic GNN representations is reserved for future work.

---

### Author Response · Authors · 2025-11-20
**General Response to Reviewers**

We sincerely thank the reviewers for their constructive feedback and for acknowledging the value of our work.

We are encouraged that all reviewers recognized the novelty and intuitive appeal of our fidelity-driven reconstruction mechanism (Reviewers 6RZd, zF7z) and agreed that our threat model is well-grounded and practically motivated (Reviewers qKuM, zF7z).

To the best of our knowledge, AFR is the first attack model designed to explicitly handle the "curse of perfection" inherent in prior spectral leakage studies. Unlike existing works that rely on idealized, noise-free assumptions, our framework leverages a Fidelity Score and Adaptive Stitching to robustly recover topology from fragmented and heterogeneous data. We believe this work bridges the critical gap between theoretical spectral leakage possibilities and practical privacy risks in Federated Graph Learning.

In the **revised manuscript**, we have incorporated significant updates to address the reviewers' suggestions regarding experimental scope and defensive countermeasures:
- Robustness against defenses (Appendix F & Section 4.3): Addressing the lack of defense evaluation (Reviewers qKuM, zF7z), we added a comprehensive study evaluating AFR against $(\epsilon, \delta)$-Gaussian Differential Privacy. We quantify the privacy-utility trade-off, demonstrating that AFR remains a potent threat (retaining high F1 scores) under moderate privacy budgets where downstream utility is preserved.
- Extended benchmarks (Section 4.2 & Table 2): We significantly expanded our evaluation to 9 diverse datasets (Reviewers 6RZd, qKuM). The inclusion of massive computer vision graphs (COCO-SP, 58M nodes) and molecular graphs (PCQM-Contact) provides strong empirical evidence for the scalability and generalizability of our method across different domains.
- We have re-run all core experiments. All results in the revised manuscript (Tables 1, 2, 6) now report **mean $\pm$ standard** deviation over 5 independent random seeds to ensure stability and reproducibility (Reviewer 6RZd).
- We also provide a detailed empirical runtime analysis for all datasets (Reviewer qKuM), demonstrating that AFR scales effectively to massive graphs (COCO-SP, 58M nodes) for offline attacks.
- Hyperparameter sensitivity (Appendix C.3): We added extended sensitivity analyses for the structural parameters $s_{min}$ and $k_{base}$ (Reviewer qKuM), confirming that the model performs robustly within a stable operating range.

We are grateful to the reviewers for their detailed and constructive feedback on our submission. We believe these revisions directly address the core concerns raised during the review process.

---

### Meta-Review · Area_Chair_dLUS · 2026-01-07

**Summary:**

This paper introduces AFR (Adaptive Fidelity-driven Reconstruction) which is a robust new attack model that abandons idealized assumptions. Experiments on multiple graph benchmarks show the proposed AFR can recover graph topology even under noisy, fragmented, and heterogeneous conditions. One weakness of this work is its complexity, e.g., the approach includes multiple distinct components and number of hyper-parameters. Other concerns raised by reviewers are the proposed method relies on sufficient patch overlap and high-fidelity core patches and more comparisons are required, which is why I recommend rejection.

**Reviewer Concerns:**

Part of concerns were addressed and discussed by the rebuttal.

**Reviewer Scores:**

They might keep initial scores.

---

### Decision · Program_Chairs · 2026-01-26

Reject